# Genetic interaction mapping reveals functional relationships between peptidoglycan endopeptidases and carboxypeptidases

**Manuela Alvarado Obando**[1,2], **Diego Rey-Varela**[3], **Felipe Cava**[3], **Tobias Dörr**[1,2,4]*

1 Department of Microbiology, Cornell University, Ithaca, New York, United States of America, 2 Weill Institute for Cell and Molecular Biology, Cornell University, Ithaca, New York, United States of America, 3 The Laboratory for Molecular Infection Medicine Sweden (MIMS), Umeå Center for Microbial Research (UCMR), Science for Life Laboratory (SciLifeLab), Department of Molecular Biology, Umeå University, Umeå, Sweden, 4 Cornell Institute for Host-Microbe Interactions and Disease (CIHMID), Ithaca, New York, United States of America

* tdoerr@cornell.edu

**Data Availability Statement:** Sequencing data were deposited to the NCBI Sequence Read Archive (SRA) under BioProject accession codes PRJNA1088767 (V. cholerae transposon-insertion

## Abstract

Peptidoglycan (PG) is the main component of the bacterial cell wall; it maintains cell shape while protecting the cell from internal osmotic pressure and external environmental challenges. PG synthesis is essential for bacterial growth and survival, and a series of PG modifications are required to allow expansion of the sacculus. Endopeptidases (EPs), for example, cleave the crosslinks between adjacent PG strands to allow the incorporation of newly synthesized PG. EPs are collectively essential for bacterial growth and must likely be carefully regulated to prevent sacculus degradation and cell death. However, EP regulation mechanisms are poorly understood. Here, we used TnSeq to uncover novel EP regulators in *Vibrio cholerae*. This screen revealed that the carboxypeptidase DacA1 (PBP5) alleviates EP toxicity. *dacA1* is essential for viability on LB medium, and this essentiality was suppressed by EP overexpression, revealing that EP toxicity both mitigates, and is mitigated by, a defect in *dacA1*. A subsequent suppressor screen to restore viability of *ΔdacA1* in LB medium identified hypomorphic mutants in the PG synthesis pathway, as well as mutations that promote EP activation. Our data thus reveal a more complex role of DacA1 in maintaining PG homeostasis than previously assumed.

## Author summary

Bacteria are surrounded by a mesh-like layer made up of a polymer known as peptidoglycan (PG). This structure provides them with shape and protection against internal osmotic pressure, which makes it essential for bacterial survival. For cells to grow and divide, PG needs to be constantly synthesized and remodelled. The enzymes involved in PG synthesis have been extensively studied; however, the enzymes and mechanisms involved in PG remodelling remain poorly understood. PG-lytic enzymes are required for

sequencing) and PRJNA1088760 (V. cholerae suppressor genome sequencing). Other datasets were deposited to Figshare and can be accessed at https://doi.org/10.6084/m9.figshare.25425721.v4.

**Funding:** This work was supported by NIH grant R01GM130971 (TD). The Cava lab (FC) was supported by The Swedish Research Council, Umeå University, the Knut and Alice Wallenberg Foundation (KAW), the Kempe Foundation and the Wenner Gren Foundation. The funders had no role in study design, data collection and analysis, decision to publish, or preparation of the manuscript.

**Competing interests:** The authors have declared that no competing interests exist.

proper growth and morphogenesis, as they perform multiple tailoring modifications that produce a mature and flexible cell wall; however, due to their potentially harmful activities (degradation of PG) their dysregulation can be catastrophic for the cell. In this study we present data that reveal a previously unknown functional relationship between two different PG-lytic enzymes (endopeptidases and carboxypeptidases), revealing a mechanism of endopeptidase regulation and a novel role of carboxypeptidases in maintaining the balance between synthesis and degradation. These findings have broad implications, offering potential for more effective antibiotics by adjusting the equilibrium between PG synthesis and degradation.

## Introduction

The bacterial cell wall serves as a vital structure for bacterial growth and survival. It fulfils multiple functions, including acting as a protective layer that imparts structural stability, facilitating adaptation to drastic environmental changes, and determining cell shape. The cell wall is composed of peptidoglycan (PG), a lattice-like structure made up of alternating N-acetyl glucosamine (GlcNAc) and N-acetylmuramic acid (MurNAc) subunits. Each MurNAc unit bears a short peptide stem composed of 5 amino acids. The peptide side stem's composition is species-specific and can also vary across growth conditions; however, in Gram-negative rods like *V. cholerae*, the most common composition is L-ala, D-glu, mDAP, D-ala, D-ala [1]. PG expansion during growth is mediated by two key synthetic processes, transglycosylation (TG) and transpeptidation (TP) reactions. TG and TP reactions are performed by PG synthases like penicillin-binding proteins (PBPs) and shape, elongation, division and sporulation (SEDS) proteins [1]. During transglycosylation, the PG strands are polymerized by forming β-[1,4] glycosidic bonds. During or after PG strand polymerization, the transpeptidation reaction is used to form crosslinks between adjacent chains via the formation of peptide bonds between most commonly the mDAP residue from one peptide stem (the "acceptor") and the 4th D-ala of another (the "donor"); this results in loss of the terminal (5th) D-ala from the donor strand (while, importantly, the acceptor strand can retain the 5th D-ala).

Besides PG synthesis, multiple tailoring modifications are required to produce a mature and flexible cell wall. These modifications are carried out by a group of enzymes collectively referred to as "autolysins", which can cleave almost every linkage in the PG structure. Autolysins include amidases, lytic transglycosylases, carboxypeptidases and endopeptidases (Fig 1A). Carboxypeptidases remove the terminal D-ala of the pentapeptide chain of newly synthesized peptidoglycan (which is where pentapeptide is expected to originate from exclusively)[2]. Their proposed function is to control the amount of pentapeptide subunits that are available for crosslinking by transpeptidation, but how exactly this contributes to proper growth and morphogenesis is unclear [3]. In *E.coli*, DacA (PBP5) is required for maintaining cell diameter, surface uniformity and overall topology of the peptidoglycan sacculus [4]; however, the mechanisms behind the morphological aberrations that occur when DacA is absent have not been elucidated [5]. In *V. cholerae*, DacA1 deletion results in drastically impaired growth and morphology in NaCl-containing medium (LB) and DacA1 is consequently (for an unknown reason) essential for viability in high-salt, but not low-salt conditions [6].

Another prominent autolysin group, the endopeptidases, cleave the crosslinks between peptide side stems of two adjacent PG strands. This activity is thought to be required for the directional insertion of new PG material during cell elongation (and probably division); consequently, EP activity is essential for growth and viability in all bacteria studied in this

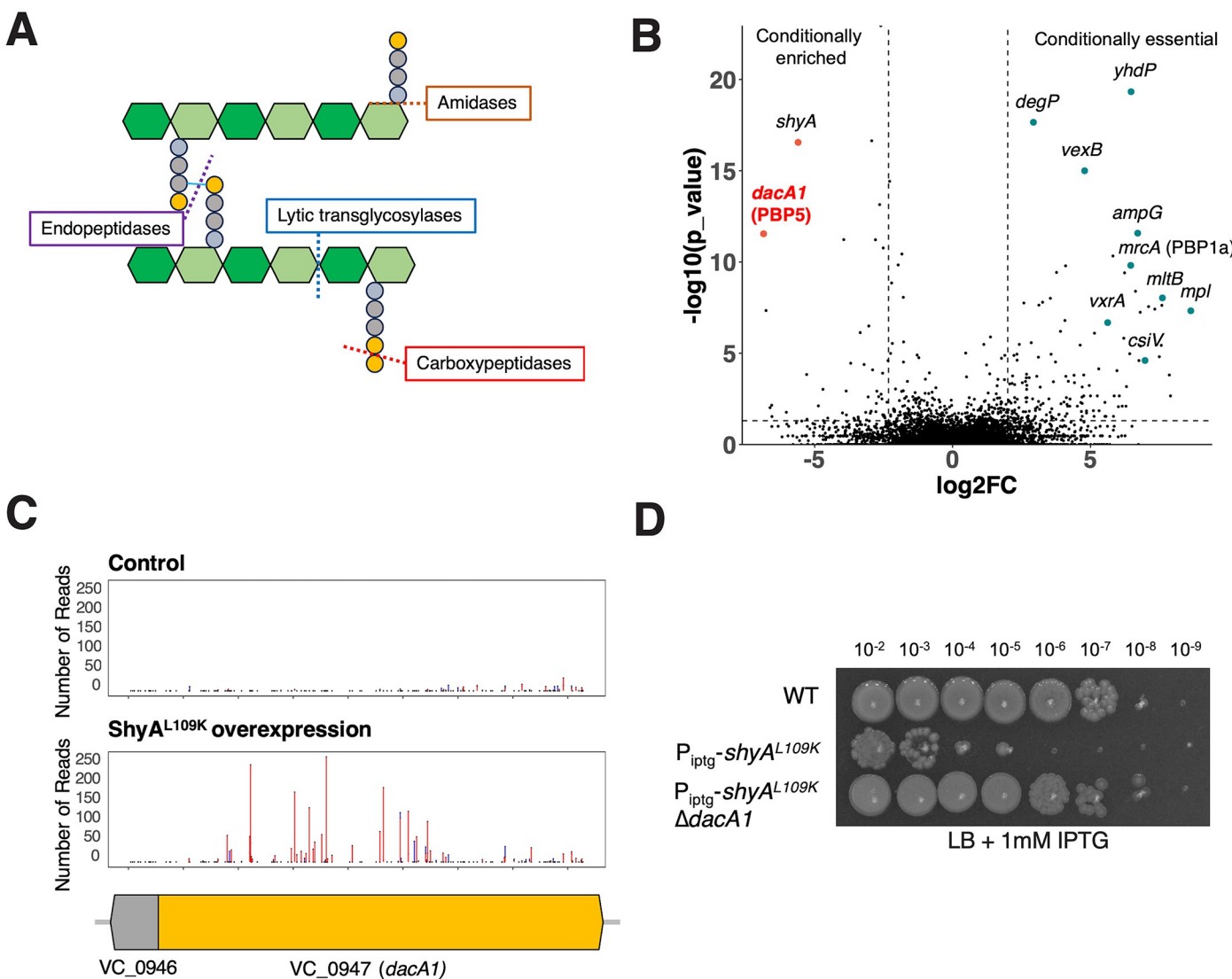

**Fig 1. A screen for EP regulators identifies a genetic relationship between ShyA and PBP5. (A)** Schematic representation of autolysin cleavage sites in the cell wall. **(B)** Volcano plot of the change in relative abundance of insertion mutants between the control condition (WT) and the experimental condition (ShyA$^{L109K}$ overexpression) (X-axis) versus p-value (Y-axis). The dashed line indicates the cutoff criterion (>2-fold fitness change, P <0.005) for identification of genes that modulate ShyA$^{L109K}$ toxicity. **(C)** *dacA1* was identified as conditionally enriched upon ShyA$^{L109K}$ overexpression using CON-ARTIST, red bars represent the number of reads from transposon insertions found in the forward orientation and blue bars represent insertions found in the reverse orientation. The black dots indicate every TA site available for transposon insertion **(D)** Overnight cultures of WT, P$_{iptg}$-shyA$^{L109K}$ and P$_{iptg}$-shyA$^{L109K}$ *ΔdacA1* where plated on LB medium with 1mM IPTG and incubated overnight at 30°C.

regard to date [7–9]. Interestingly, EPs represent a double-edged sword for the cell, as they are at once essential for cell growth, but also potentially detrimental to the structural integrity of the PG sacculus if not properly regulated. Tight regulation of endopeptidase activity at an optimal level is therefore likely essential to maintain cell growth and morphology. Endopeptidase regulation mechanisms have only begun to be understood in Gram-negative bacteria; such mechanisms include regulation at the transcriptional level [10], proteolytic degradation [11], substrate selectivity [7] and post-translational activation through conformational changes [12]. The latter mechanism is an important aspect of endopeptidase regulation in *V. cholerae's* main endopeptidase ShyA, where conformational re-arrangements allow ShyA to transition from a

closed (inactive) conformation (where domain 1 occludes the catalytic groove of domain 3), to an open (active) conformation where this inhibition is relieved [12]. The factors involved in this conformational switching remain unknown. Mutating residues in the domain 1/domain 3 interaction surface (most prominently the L109K change) causes disruption of hydrophobic interactions between domain 1 and 3, resulting in a more open protein conformation and subsequent hyperactivity [12]. How EP activity is harnessed to properly direct PG insertion during cell elongation, and how EPs intersect with other autolysins, is largely unknown; crucially, all the regulatory mechanisms outlined above are not essential for growth (while endopeptidases are), suggesting additional regulatory pathways.

Here, we reveal a new functional interaction between carboxypeptidases and EPs. We employed TnSeq to identify factors that exacerbate or mitigate toxicity of an activated EP and discovered the carboxypeptidase DacA1(PBP5) as a previously unknown alleviator of cell wall degradation. Importantly, *dacA1* itself is essential for growth, and this was suppressed by EP overexpression. A screen for additional suppressors of *dacA1* essentiality revealed mutants in the PG synthesis pathway that exhibit decreased function, as well as mutations that promote EP activation. Our findings thus suggest a key role for DacA1 in preserving the equilibrium between PG synthesis and degradation.

## Results

### A screen for endopeptidase regulators identifies a genetic relationship between ShyA and DacA1 (PBP5)

As part of an ongoing effort to identify regulators of peptidoglycan endopeptidase activity, we conducted a Transposon Sequencing Screen (TnSeq) in *V. cholerae* to identify genes that modulate the toxicity of the activated version of the EP ShyA (ShyA$^{L109K}$). Overexpression of ShyA$^{L109K}$ causes enhanced cleavage activity due to the mutational relief of an intramolecular inhibitory mechanism [12]. We created Tn insertion libraries (3 x 200,000 colonies) in a strain background that overexpresses ShyA$^{L109K}$ from a chromosomal locus and identified genome-wide Tn insertion frequencies vs. WT control. The screen identified several Tn insertion events that were significantly underrepresented in the ShyA$^{L109K}$ strain (indicating conditional essentiality in this background), as well as some that were significantly overrepresented, indicating genes whose inactivation reduces ShyA$^{L109K}$ toxicity (S1 Table).

Among the conditionally essential genes, this screen revealed PG synthesis and recycling factors such as the main PBP, *pbp1A*, its accessory factor *csiV* [13], and several PG recycling pathway components, namely the muropeptide ligase *mpl*, the muropeptide fragment permease *ampG* [14] and the lytic transglycosylase *mltB* (Fig 1B). Another synthetic lethal hit was *yhdP*, which likely fulfils phospholipid transport functions to the OM, with a preference for saturated fatty acids [15]. ShyA$^{L109K}$ overexpression causes spheroplast formation, and spheroplast survival has been shown to depend on YhdP [16]; this hit may thus suggest that saturated fatty acids contribute to spheroplast stability. As internal validation, insertions in *shyA* were identified as synthetic healthy, likely reflecting insertions into the chromosomal ShyA$^{L109K}$ overexpression construct (Fig 1B). Intriguingly, the gene encoding the carboxypeptidase DacA1 was conditionally enriched in our TnSeq, indicating a synthetic healthy relationship between *dacA1* and *shyA$^{L109K}$* (Fig 1B and 1C). The deletion of *dacA1* in cells overexpressing ShyA$^{L109K}$ resulted in complete rescue of cell growth, validating the TnSeq results (Fig 1D). This indicates that DacA1, the main carboxypeptidase in *V. cholerae*, either directly or indirectly exacerbates toxicity of ShyA$^{L109K}$, suggesting an uncharacterized link between CPase and EP activity. We were intrigued by this genetic relationship and chose to characterize this connection further.

## PG of *ΔdacA1* mutant is less susceptible to ShyA cleavage

We have previously shown that ShyA exhibits substrate selectivity, i.e. it prefers cleaving cross-links that contain tetrapeptides over those that contain pentapeptides, and speculated that this represents a strategy to differentiate between old and new PG (7). We have also shown that DacA1 is the main carboxypeptidase under normal laboratory conditions, as its deletion results in a >8-fold increase in pentapeptide content [6]. We thus hypothesized that the increase in pentapeptide content in the absence of DacA1 causes reduced susceptibility to ShyA[L109K] cleavage, thus conferring resistance against EP toxicity (Fig 2A). To test this hypothesis, we purified ShyA and ShyA[L109K], and used a Remazol Brilliant Blue PG degradation assay to quantify PG hydrolysis. As substrate, we used PG purified from either WT or *ΔdacA1* cells (Fig 2B). The activity of both ShyA and ShyA[L109K] was significantly reduced on *ΔdacA1* PG compared to WT PG, suggesting that indeed *ΔdacA1* cell wall is at least partially refractory to EP degradation. To exclude the possibility that *ΔdacA1* PG is somehow generally less susceptible to enzymatic degradation (through, for example, non-specific PG architectural traits), we also included lysozyme as a control, which should be active regardless of pentapeptide content due to its activity on PG's polysaccharide backbone. Indeed, lysozyme activity was unaffected by PG origin (Fig 2C). These results indicate that modifications in the PG side stem structure generated by DacA1 (likely pentapeptide) reduce ShyA activity, suggesting a role for DacA1 in substrate-guided endopeptidase regulation.

**Upregulation of cell wall degradation functions suppresses ΔdacA1 essentiality.** The finding that a *dacA1* mutant rescued ShyA[L109K] overexpression toxicity was unexpected, since DacA1 itself is essential for growth in LB (the growth medium in which we conducted our screen), but non-essential for growth in salt-free LB [6]. The absence of DacA1 in *V. cholerae* results in severe growth and morphological defects during growth in LB medium, and these phenotypes specifically depend on the NaCl concentration, rather than osmolarity [6]. In contrast, *ΔdacA1* cells did not present growth defects or aberrant morphologies when grown in salt-free LB, though *ΔdacA1* cells were significantly longer and wider when compared to WT (S1C Fig). When validating the genetic relationship between ShyA[L109K] and DacA1, we found that increased endopeptidase activity from ShyA[L109K] rescued both the *ΔdacA1* growth and morphology defects in LB (Fig 3) (S1A and S1B Fig). Thus, curiously, the *ΔdacA1* mutation and ShyA[L109K] overexpression are two individually toxic events that suppress each other's detrimental effects on the cell. This observation suggests that reduction in PG degradation might contribute to *ΔdacA1* defects in LB. To address the genetic basis of the *ΔdacA1* salt sensitivity phenotype, and thus potentially further characterize its relationship with ShyA, we performed three independent genetic screens to select for suppressor mutations that rescued the *ΔdacA1* mutant's growth on LB medium and mapped them via whole genome sequencing (S2 Table). A common pattern in the mutations that were selected for was an enrichment in mutants associated with either PG degradation, or PG synthesis, elaborated in further detail below.

Among the genes that are associated with PG degradation, we found suppressor mutations in two genes that are expected to result in endopeptidase activation. One mutation mapped to *zur*, a Fur family transcriptional regulator known to repress the expression of ShyB—a ShyA paralog—under zinc replete conditions [10] (Fig 4A). We hypothesized that the *zur* mutation represented a loss of function allele, resulting in upregulation of ShyB. To test this, we constructed a clean deletion of *zur* in a *ΔdacA1* background. This resulted in partial rescue of the *ΔdacA1* phenotype in LB (Fig 4A). We then confirmed that partial rescue was due to *zur* deletion and not a polar effect, by expressing *zur in trans*. Next, we constructed a strain where ShyB was under the control of an IPTG inducible promoter to test more directly its ability to suppress *ΔdacA1* growth defects. Indeed, ShyB expression rescued *ΔdacA1* normal growth

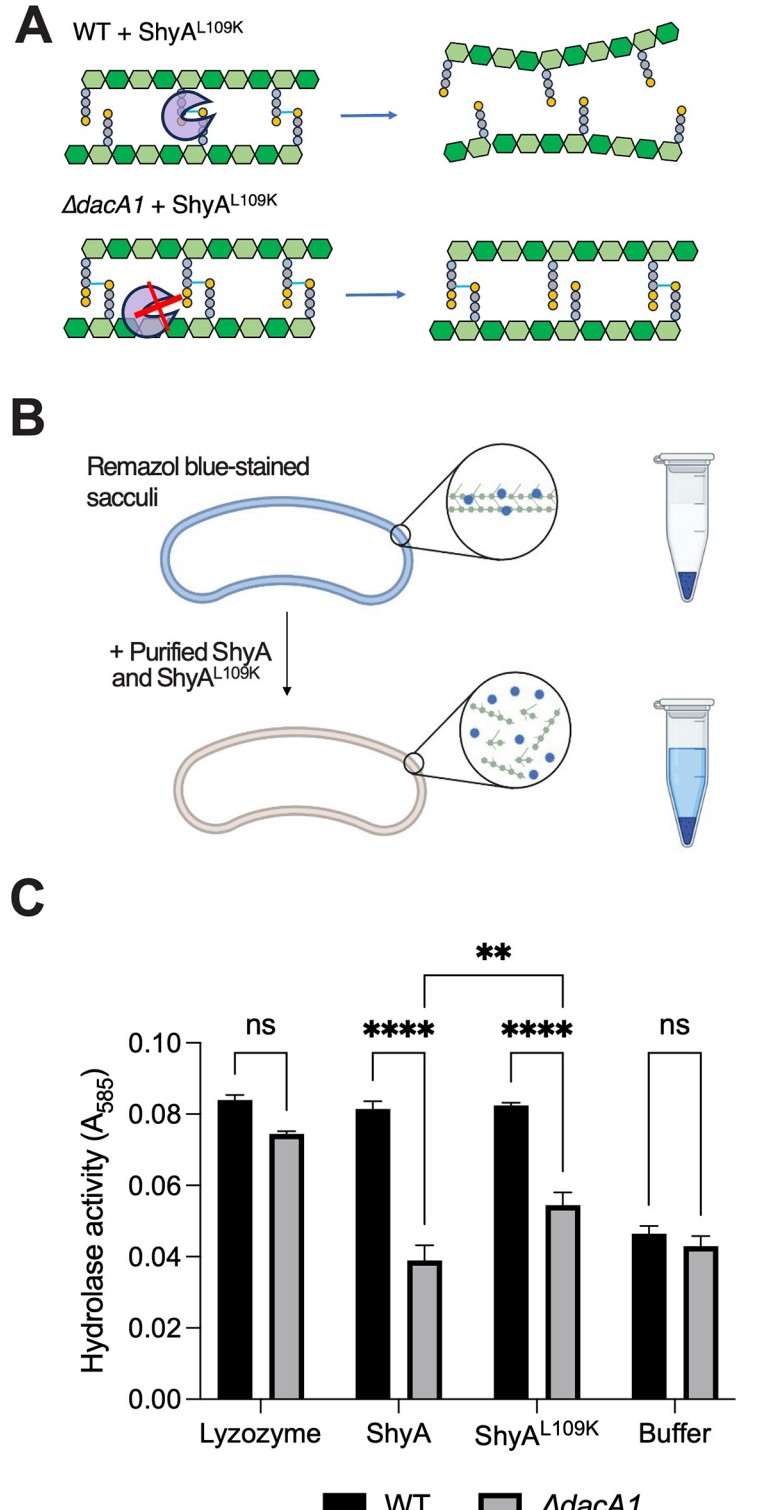

**Fig 2. ΔdacA1 mutant PG is less susceptible to cleavage by ShyA. (A)** Schematic representation of how DacA1 absence affects the pentapeptide content in the PG sacculus and potentially ShyA activity. **(B)** Schematic of experimental procedure. The PG sacculi from WT and ΔdacA1 cells were purified using the SDS precipitation method [52]. Purified samples were then stained with Remazol Brilliant Blue and used in sedimentation assays where PG fragmented by hydrolases remains soluble in the supernatant upon centrifugation, providing a quantitative readout of EP activity. Figure was made with Biorender **(C)** RBB stained sacculi were treated as described in (B) with the addition

of proteins indicated on the x-axis. Undigested material was removed by centrifugation and the supernatant was used to measure OD$_{585}$. Error bars represent standard deviation of two independent experiments. Significance was determined using a two-way ANOVA, **** corresponds to a P-value of <0.0001.

(Fig 4B) as well as morphological defects (Fig 4E) (S2 Fig) in LB medium, showing that upregulation of another EP can suppress Δ*dacA1* essentiality.

Most intriguingly, the screen also revealed a mutation in ShyA (R115W). Residue R115 lies in the interaction surface between ShyA's inhibitory domain 1 and its active site domain 3 (Fig 4D) close to many residues we have previously shown to activate ShyA activity [12]. Consistent with an activating role of this mutation, ShyA$^{R115W}$ exhibited increased toxicity (reduced plating efficiency and morphological defects) when expressed in *E. coli* (S4 Fig). We then overexpressed ShyA$^{R115W}$ from an IPTG inducible promoter in the WT and Δ*dacA1* backgrounds (and validated their expression via Western Blot (S1D Fig)). Unlike ShyA$^{L109K}$, overexpression of ShyA$^{R115W}$ did not affect growth or produce spheroplasts in a *V. cholerae* WT background; however, it was (unlike wt ShyA) still able to partially rescue growth and morphology in the Δ*dacA1* phenotype in LB medium (Fig 4C and 4E) (S2 Fig).

*V. cholerae* encodes 6 D,D-endopeptidases, two of which (ShyB and activated ShyA) are able to rescue the Δ*dacA1* phenotype in LB. We wondered if the remaining 4 D,D endopeptidases were able to rescue the Δ*dacA1* phenotype as well, since they perform the same enzymatic reaction. To this end, we overexpressed NlpC, TagE1 (*vc0843*), TagE2 (*vca1043*), and ShyC in the Δ*dacA1* background and found that overexpression of TagE2 and ShyC also alleviated the morphological defects caused by *dacA1* deletion (S3 Fig). Taken together, these results indicate that increased endopeptidase activity is beneficial for the Δ*dacA1* mutant in LB medium.

One possible explanation for EPs' rescue effect is that activated ShyA and ShyB have carboxypeptidase activity. While our extensive previous *in vitro* biochemistry work characterizing these enzymes [10,12] did not reveal any such activity, we turned to *in vivo* PG analysis to validate the previous *in vitro* results. To this end, we purified sacculi from WT, Δ*dacA1* and Δ*dacA1* overexpressing either ShyA, ShyB, or ShyA$^{R115W}$ and analyzed them by LC/MS (see Methods for details). While overall pentapeptide content was reduced in the EP overexpression strains (between 87 and 92% of Δ*dacA1* pentapeptide), it was still dramatically higher than wt levels (11% of Δ*dacA1* pentapeptide) (S3 Table). Most importantly, pentapeptide of the ShyA overexpression strain (which does not rescue Δ*dacA1*) was equal to that of the ShyB overexpression strain (which does rescue), demonstrating that the reduction in pentapeptide content observed in these strains is not sufficient for mitigation of Δ*dacA1* growth defects. In addition, M5 monomer content was similar between all strains (at 10fold higher than WT), further supporting lack of significant carboxypeptidase activity of these EPs. Interestingly, however, while PG dimers containing only tetrapeptide (D44) were unaffected by EP overexpression in the Δ*dacA1* background, D45 (crosslinked PG subunits containing tetrapeptide in one stem, and pentapeptide in the other) content was reduced significantly, suggesting that activated EPs can cleave pentapeptide-containing crosslinks at least to a low degree (S3). Thus, activated EPs likely suppress *dacA1* phenotypes by circumventing ShyA's inability to cleave pentapeptide-containing crosslinks.

**Downregulation of cell wall synthesis suppresses ΔdacA1 essentiality.** The second category of mutants identified by the screen mapped to genes involved in PG precursor synthesis (*murA*$^{P122S}$, *murA*$^{L35F}$ *murC*$^{A132T}$ and *murD*$^{D447E}$). Since *murA*, *murC* and *murD* are essential for growth, these mutations must represent either hypomorphs or gain-of-function mutations,

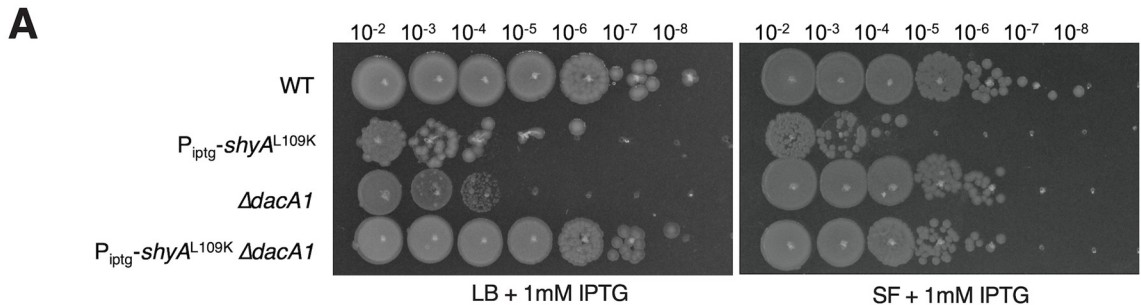

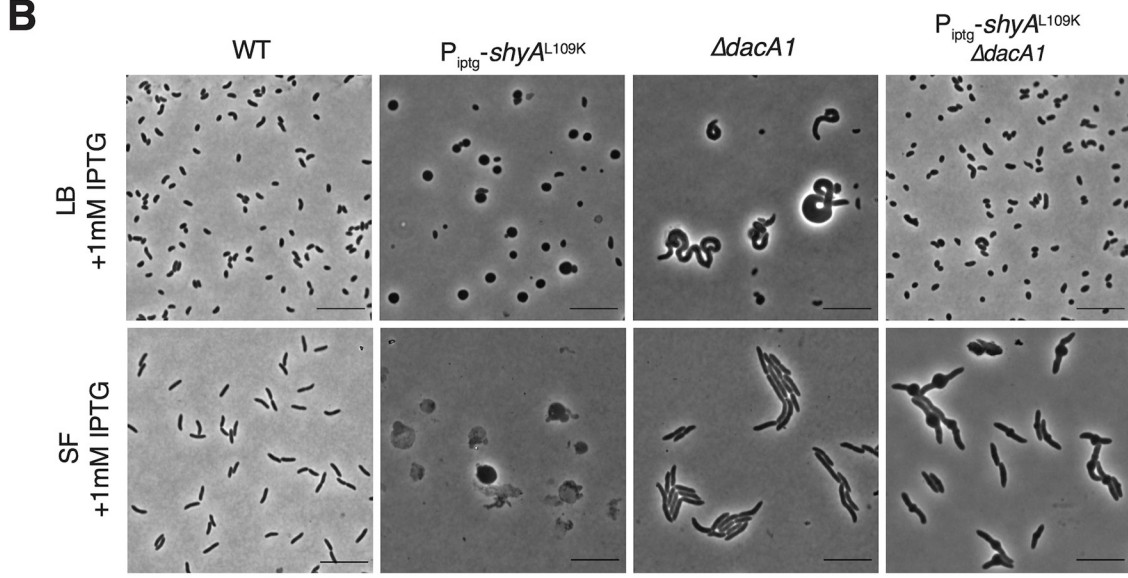

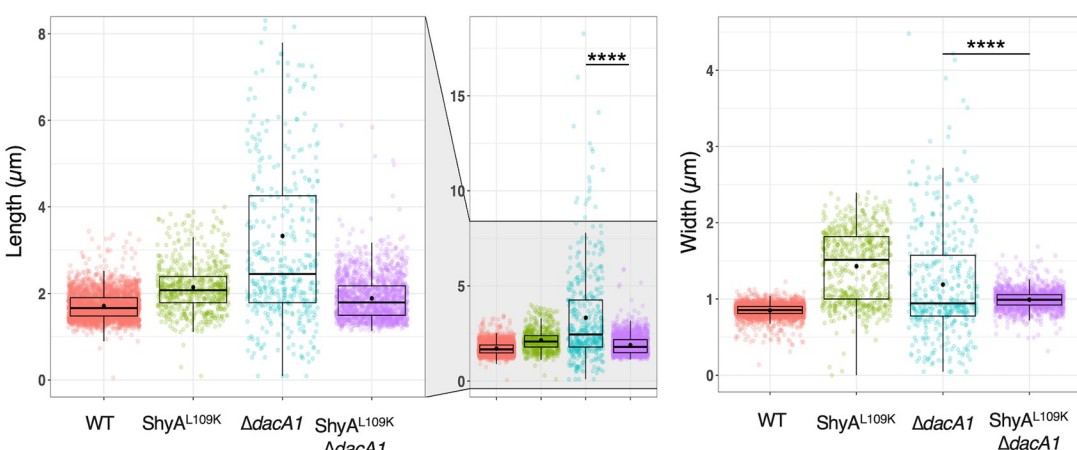

**Fig 3. ShyA^{L109K} overexpression restores normal cell growth and morphology in Δ*dacA1* cells grown on LB medium. (A)** Overnight cultures were grown on LB (WT background) and salt-free LB (Δ*dacA1* background) and then spot-titered on either LB or salt-free LB (SF) with 1mM IPTG. **(B)** Overnight cultures were sub-cultured 1:100 in either LB or SF at 37˚C until they reached exponential phase, followed by addition of inducer (1mM IPTG). Phase contrast images were taken after 2 hours of induction. Scale bar: 10 μm **(C)** Cells were segmented using Omnipose, and area, width and length were calculated with MicrobeJ. Statistical significance was determined with Welch's two sample t-test. ****, P < 0.0001. Shown are boxplots with raw data points.

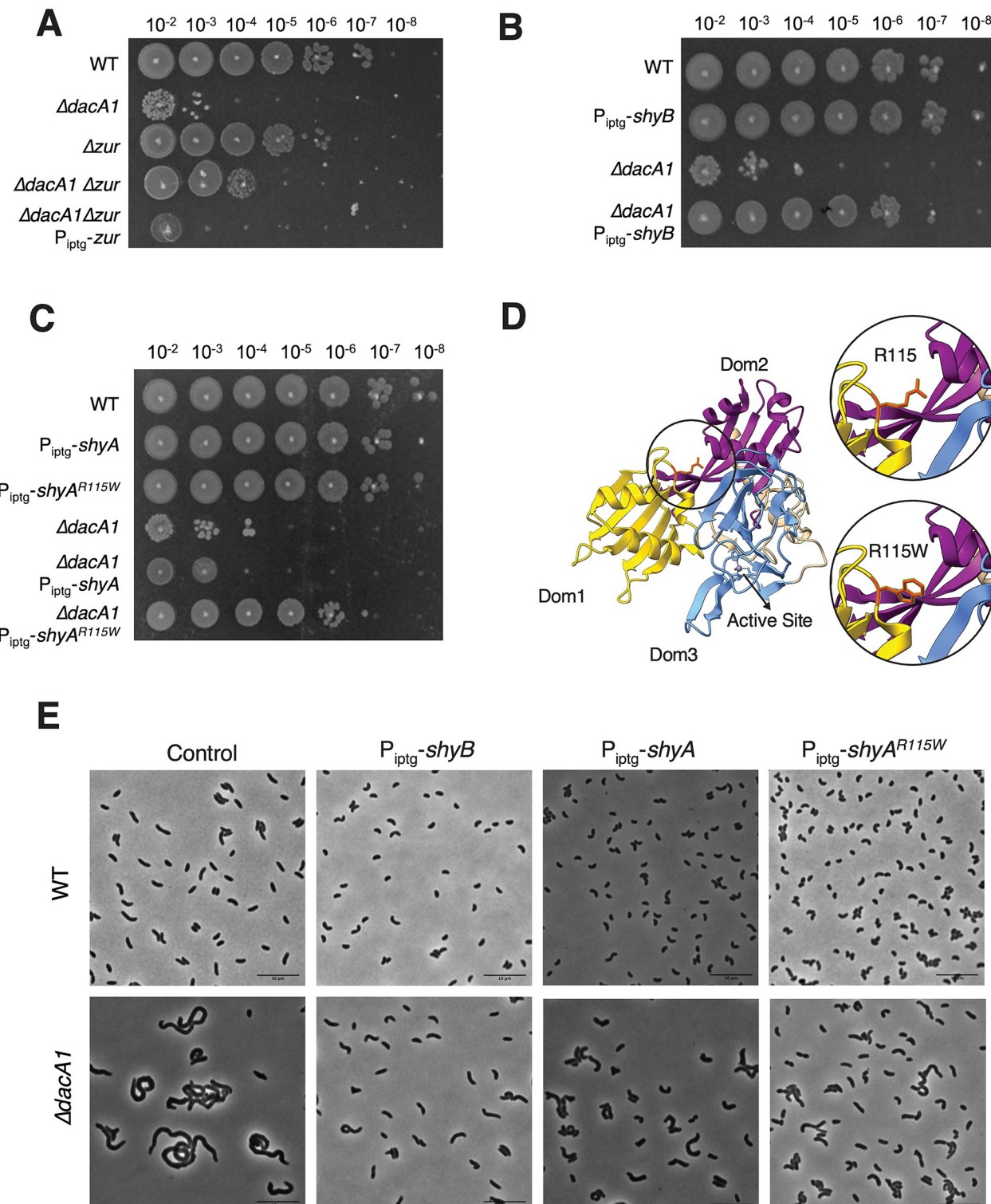

**Fig 4. Mutations that enhance endopeptidase activity suppress the *ΔdacA1* phenotype in LB.** **(A)** Overnight cultures were grown on LB (WT background) and salt-free LB (*ΔdacA1* background) and then spot-titered on LB medium with 1mM IPTG and incubated at 30˚C overnight. **(B,C)** Overnight cultures of the specified strains were diluted 1:100 in LB medium and grown at 37˚C until reaching exponential phase. Following this, 1mM IPTG was added to induce gene expression, and the cultures were further incubated at 37˚C for 3 hours. **(D)** Crystal structure of ShyA showing residue R115 in the interphase between Domain 1 and Domain 3, and prediction of the R115W mutation. **(E)** Overnight cultures were sub-cultured 1:100 in LB at 37˚C until they reached exponential phase, followed by addition of inducer (1mM IPTG). Phase contrast images were taken after 3 hours of induction. Scale bar: 10 μm.

rather than loss of function. Based on our finding that increased PG hydrolysis rescued the Δ*dacA1* mutant, we hypothesized that these *mur* mutations may accomplish something similar by reducing PG synthesis, i.e. that they are hypomorphs. The MurA structure consists of two globular domains, with its active site located in the interface between them [17]. In *E. coli*, residue C115 has been identified as the catalytic cysteine required for MurA activity, which is often targeted by MurA inhibitors [18]. Residue P122, which was mutated to serine in one of our suppressors, is situated in the interface between both domains and near the active site. Proline is the most rigid amino acid [19], therefore replacing it with a serine (P122S) will likely induce a significant conformational change affecting MurA activity. The other mutation (L35F) is located in the N-terminal domain of the protein, and its effect is thus harder to gauge; however, reports in other organisms suggest mutations in the globular domains may have a role in MurA turnover by ClpCP [20] and protein-protein interactions [21]. Mur ligases MurC and MurD contain three functional domains, where Domain 1 accounts for the binding of the UDP moiety of the UDP-sugar substrate, Domain 2 binds ATP via the ATP-binding consensus sequence (GKT) and Domain 3 binds the amino acid substrate. In order for these enzymes to form a functional active site, they must undergo structural rearrangements once the substrates are bound [22]. The mutation in MurC (A132T) is located near the GKT domain, which could result in hindered ATP binding and decreased MurC activity. On the other hand, the mutation in MurD (D447E) is located in Domain 3 which could result in reduced binding of the amino acid substrate, or the prevention of conformational rearrangements required for proper enzyme function (S5 Fig).

To test the role of these mutations in suppressing Δ*dacA1* defects further, we first reconstituted the *murA$^{P122S/L35F}$* and *murD$^{D447E}$* mutations in naïve WT and Δ*dacA1* backgrounds (we were unable to construct *murA$^{P122S}$* in the wild-type background, and murC$^{A132T}$ in either background, likely due to growth-prohibitive reduction in function of these alleles. All three reconstituted mutations restored colony formation of Δ*dacA1* on LB (Fig 5A) and conferred a (albeit subtle) growth advantage in LB, which is more evident during exponential phase (S7A Fig). Further investigation revealed some complexity. For example, neither mutation restored the morphological defects; rather, they exacerbated the division defect. Δ*dacA1* cells with mutations *murA$^{P122S/L35F}$* and *murD$^{D447E}$* cells were 65%, 39% and 30% larger, and 92%, 72% and 9% longer, respectively. However, these MurA mutants exhibited ~15% decrease in width compared to the Δ*dacA1* mutant, while the *murD$^{D447E}$* mutant exhibited a 7% increase in width (Fig 5B) (S6 Fig). Thus, these mutations uncouple morphological aberrations from the ability of Δ*dacA1* to ultimately form a colony on a plate. Interestingly, mutations *murA$^{P122S}$* and *murD$^{D447E}$* made Δ*dacA1* sensitive to salt-free LB, even though they rescued the Δ*dacA1* growth defect on LB medium, suggesting fine-tuned optimality in these mutations' ability to rescue Δ*dacA1*. In addition to morphological and growth assessments, we tested sensitivity of the *murA* mutants to fosfomycin, an antibiotic that inhibits MurA function, with the rationale that reduced MurA activity would make these mutants more susceptible to this antibiotic. We found that in the WT and Δ*dacA1* backgrounds *murA$^{L35F}$* exhibited increased sensitivity to fosfomycin, and so did *murA$^{P122S}$* in the Δ*dacA1* background (S7B Fig), solidifying this hypothesis.

We next turned to an *in vitro* assay to biochemically probe *mur* mutant function. We purified MurA, MurA$^{P122S}$, MurA$^{L35F}$, MurC and MurC$^{A132T}$ and measured their enzymatic activity using MurA and MurC assay kits, respectively [23]. The MurA mutants (MurA$^{P122S}$ and MurA$^{L35F}$) exhibited significantly lower activity than the WT, almost comparable to MurA treated with its inhibitor, fosfomycin (Fig 5C). Similarly, MurC$^{A132T}$ exhibited lower enzymatic activity when compared to WT MurC (Fig 5D). These results suggested that reducing cell wall synthesis alleviates Δ*dacA1* growth phenotypes. To further test this, we purified sacculi from

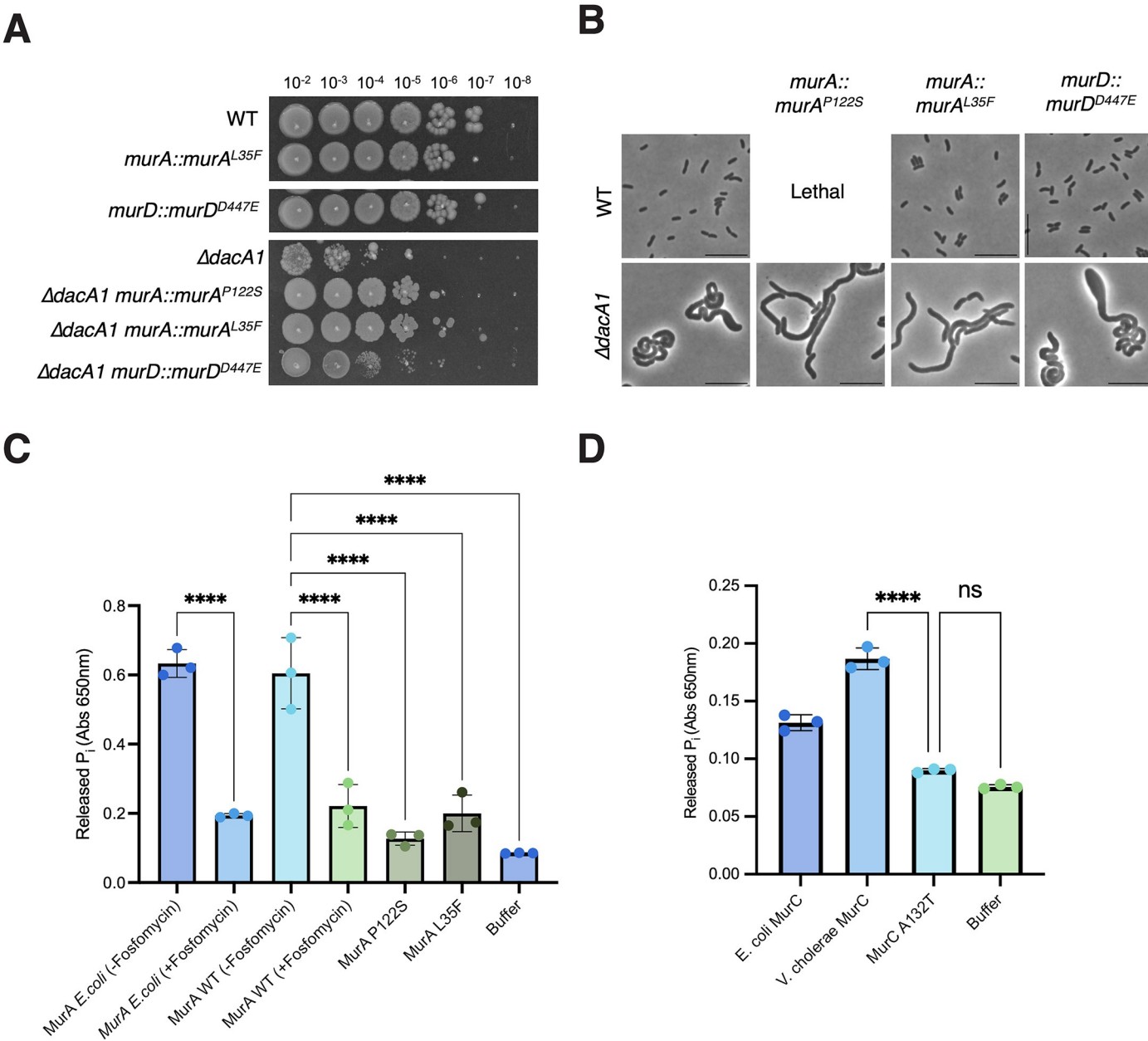

**Fig 5. Hypomorphic mutants in the PG precursor synthesis pathway rescue ΔdacA1 growth but not morphology. (A)** Indicated strains are chromosomal replacements of *murA* and *murD* with *murA^P122S^*, *murA^L35F^* and *murD^D447E^*, respectively. Overnight cultures were plated on LB medium by spot- titering and incubated at 30°C overnight. **(B)** Overnight cultures of the indicated strains were sub cultured 1:100 and grown at 37°C in LB medium for 1 hour, then back diluted 1:100 in LB and incubated at 37°C for an additional 2 hours before imaging **(C)** MurA as well as the different mutants were purified, and their activity was measured *in vitro* by measuring the inorganic phosphate released from the MurA enzymatic reaction. **(D)** MurC and MurC^A132T^ were purified and their enzymatic activity was quantified *in vitro* by measuring the inorganic phosphate released from the MurC enzymatic reaction. (C-D) Statistical significance was determined by one-way ANOVA, error bars represent the standard deviation of three replicates (raw data points also shown).

the *ΔdacA1* mutant and *ΔdacA1 murA::murA^L35F^* and analyzed them by LC/MS. The mutation in *murA* resulted in a significant decrease in the relative PG amount, which usually correlates with decreased PG synthesis [24] (S8 Fig). These results further corroborate the hypothesis that the rescue of the *ΔdacA1* mutant by the hypomorphic mutations in the *mur* genes is due

to a reduction in PG synthesis. If this is true, we would expect externally-induced reduction of PG synthesis to likewise result in rescued growth of ΔdacA1. To test this, we chemically reduced precursor synthesis and measured growth of ΔdacA1 cells on LB. Treatment of the ΔdacA1 mutant with increasing concentrations of fosfomycin partially restored ΔdacA1 cell growth on LB at concentrations between 5μg/mL and 15μg/mL (0.15 x—0.45 x MIC) (S9 Fig). In aggregate, we conclude that mutations in MurA, MurC and MurD promote proper growth (colony formation), but not proper morphogenesis in ΔdacA1 via PG synthesis reduction.

Depending on affinities of the multiple cell wall synthases for lipid II, reduced precursor synthesis may disproportionately affect the Rod system, divisome, and/or aPBPs. To test whether the observed fosfomycin rescue affect was due to selective reduction in the activity of one of these synthases, we treated WT and ΔdacA1 cells with sub-MIC concentrations of rod complex inhibitors (A22, MP265 and mecillinam), the aPBP inhibitor moenomycin or the PBP3 inhibitor aztreonam. We found that inhibiting the different PG synthases independently did not result in ΔdacA1 rescue (S10 Fig). To next test if generalized PG synthesis inhibition resulted in ΔdacA1 rescue, we treated WT and ΔdacA1 cells with sub-MIC concentrations of A22 and Moenomycin simultaneously. Consistent with our prediction, preventing generalized PG synthesis partially rescued ΔdacA1 in LB (S11 Fig) in a very subtle and concentration-dependent, but reproducible, way. These data suggest that the dacA1 mutant's defects are caused at least partially by surplus PG synthesis.

**ΔdacA1 salt sensitivity is partially rescued by interfering with C55 metabolism, and exacerbated by increased PG synthesis.** Our data suggest that either reducing cell wall synthesis or increasing cell wall degradation can rescue growth of the dacA1 mutant. Why then is the mutant salt-sensitive in the first place? In addition to a proton motive force for energy generation, *V. cholerae* also generates a sodium motive force to power its flagellum, and, as recently suggested, to drive the essential recycling reaction of the undecaprenol pyrophosphate cell wall precursor carrier [25]. Thus, growth in salt-free LB might limit *V. cholerae's* capacity for optimal cell wall synthesis due to the absence of a sodium motive force; conversely, growth in LB might enhance PG synthesis. If this was the case, and enhanced PG synthesis in LB was the culprit for ΔdacA1 defects, we would expect a mutant in the sodium-dependent undecaprenol pyrophosphate translocase, Vca0040, to suppress ΔdacA1 phenotypes. To test whether there was a genetic relationship between *vca0040* and *dacA1*, we created a double mutant and tested its growth on LB by spot plating, and via growth curve analysis as well as its morphology via microscopy. Consistent with our hypothesis, we observed that deleting *vca0040* in a ΔdacA1 background partially restored growth and morphology, and complementation with an IPTG inducible copy of *vca0040* demonstrated that this was in fact due to the absence of the translocase and not a polar effect (Fig 6A) (S12 Fig). These findings suggest that Vca0040 does contribute to the NaCl dependent toxicity in the ΔdacA1 mutant; however, it is not the only factor involved.

Next, we tested whether there is a relationship between NaCl toxicity and the availability of PG precursors. To this end, we overexpressed MurA (which results in upregulation of PG synthesis)[14] in a ΔdacA1 background and evaluated its effect of growth at different NaCl concentrations. We found that overexpression of MurA in ΔdacA1 cells made them more susceptible to lower concentrations of NaCl (120mM) when compared to ΔdacA1 cells not overexpressing MurA (which become susceptible to NaCl at a concentration of 140mM only) (Fig 6B) (S13 Fig). This again subtle, but reproducible phenotype indicates that increased availability of PG precursors, in addition to Na+ in the culture medium can exacerbate the growth defects observed in ΔdacA1 cells.

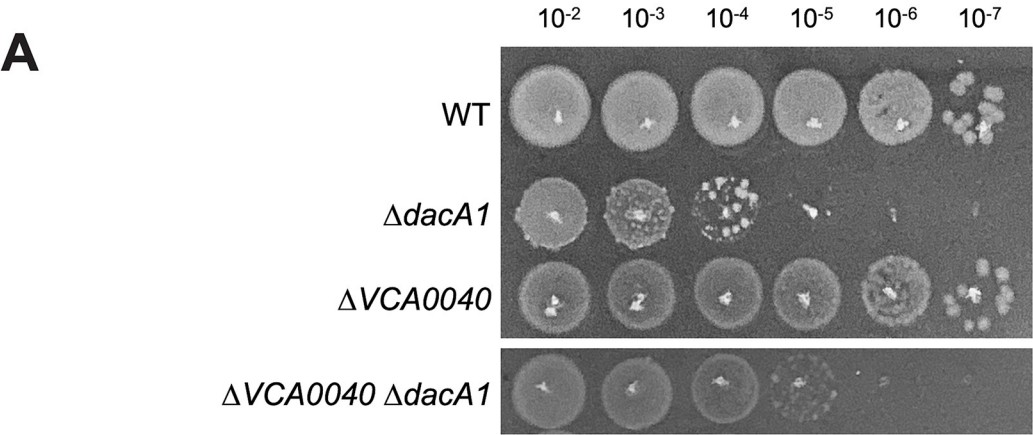

**Fig 6. PG precursor recycling and synthesis modulate *dacA1* mutant fitness. (A)** Overnight cultures of the indicated mutants were grown on LB or salt-free LB (*ΔdacA1* background) and plated on LB medium via spot-titering and incubated at 30˚C overnight. **(B)** Overnight cultures of WT, *ΔdacA1* and both backgrounds harbouring an IPTG inducible copy of *murA* were plated on salt-free LB with increasing concentrations of NaCl (120mM, 140mM and 160mM) and incubated at 30˚C overnight.

### The relationship between PBP5 and endopeptidases is conserved in *E. coli*

DacA1 homologs are collectively essential for proper cell shape maintenance in *E.coli*, and their absence leads to aberrant and irregular morphologies, similar to the *V. cholerae dacA1* mutant [4,26,27]. To investigate if increased endopeptidase activity rescues the morphological defects caused by carboxypeptidase deficiency in *E. coli*, we overexpressed ShyA and ShyA$^{R115W}$ in a strain lacking the 4 DD-CPases DacA, PbpG, DacB and DacD (Strain CS446-1 [26]), hereafter referred to as Δ4. The Δ4 strain exhibits well-characterized, severe morphological defects, quantifiable as a drastic increase in cell area as well as a slight plating defect [28]. These defects could be almost fully suppressed by overexpressing ShyA, which was achieved even with leaky expression in the presence of glucose (Fig 7) (S4 Fig). We also attempted to rescue Δ4 by overexpressing our mutant ShyA$^{R115W}$, however, this resulted in severe growth defects in both WT and *Δ4* (S4 Fig). To test if overexpression of other endopeptidases rescued the morphological defects of Δ4, we overexpressed ShyB, which was able to rescue *ΔdacA1* in *V. cholerae*, as well as a well-known ShyA homolog in *E. coli*, MepM. To our surprise, overexpression of ShyB resulted in filamentation in both the WT and Δ4 backgrounds. This phenotype is consistent with previous reports that suggest increased endopeptidase activity antagonizes cell division [29]. MepM was expressed both in its wild-type (WT) form and an active form which lacks Domain 1 (MepM Δdom1) [12]. Overexpression of WT MepM in the Δ4 strain did not alleviate the morphological defects, but overexpression of MepM Δdom1 did, which suggests that MepM, just like ShyA in *V. cholerae*, needs to be activated to rescue the defects caused by carboxypeptidase deficiency. Overexpression of MepM Δdom1 proved to be toxic in both WT and Δ4; however, leaky expression when grown in glucose was sufficient to rescue the morphological defects as well as the slight growth defect, which we also observed with ShyB (S14 Fig). Taken together, these results demonstrate that morphological defects associated with carboxypeptidase deficiency in *E. coli* can be rescued by increasing endopeptidase activity, indicating that the role of endopeptidases in rescuing defects associated with carboxypeptidase deficiency is conserved.

## Discussion

In this study, we have shown that defects associated with carboxypeptidase deficiency can be alleviated by either enhancing cell wall degradation, or by reducing cell wall synthesis. Fundamentally, this suggests that a key role of carboxypeptidases is to ensure proper balance between PG synthesis and degradation (Fig 8). Our data add support to a model that increased cell wall synthesis can be detrimental if not properly balanced by degradation. Studies in *Bacillus subtillis* described a similar phenomenon, where accumulation of PG precursors disturbed the balance between PG synthesis and degradation, which could be restored by increased D,L endopeptidase activity (LytE) or by mutations that downregulated PBP1 [30]. It is important to note, however, that increased endopeptidase activity seems to be the superior way to rescue *ΔdacA1* defects–while enhancing EP activity almost completely suppressed *ΔdacA1* growth and morphology, reduction of synthesis either chemically or through *mur* hypomorphs only partially rescued growth, but did not restore morphology. This may either indicate that more severe trade-offs are associated with downregulating synthesis than with upregulating degradation, or that ultimately, crosslink cleavage is required, and reducing synthesis indirectly allows effective cleavage of fewer crosslinks via other EPs (the *V. cholerae* genome encodes 9 such enzymes) with lower activity.

Carboxypeptidase function has been linked to cell shape maintenance in *E. coli* based on the observation that their absence results in severe morphological defects [4,27,28]. This has been attributed to excessive transpeptidation and incorrect crosslinking as a consequence of

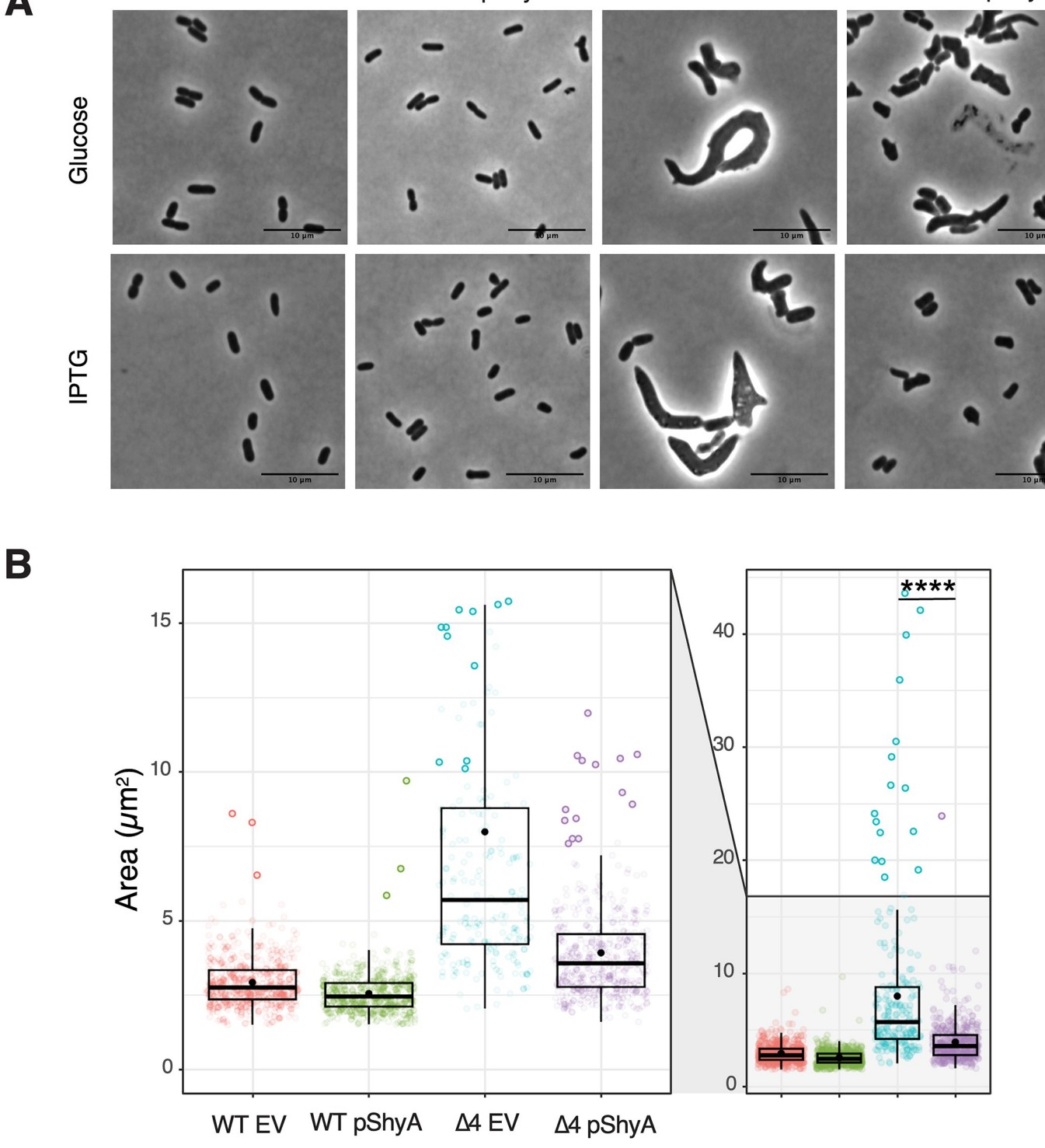

**Fig 7. ShyA overexpression rescues morphological defects associated with carboxypeptidase insufficiency in *E. coli*.** (A) *E. coli* strains CS109 and Δ4 (CS109 *ΔdacA ΔdacB ΔdacC ΔpbpG*) containing either an empty vector (pHL100) or a vector containing an IPTG-inducible copy of ShyA (pHL100-ShyA) were incubated overnight at 37˚C in LB medium with 0.2% glucose, sub-cultured 1:100 in LB and incubated for 2 hours, then back-diluted 1:100 in LB and grown for an additional 2 hours. 1 mM IPTG or 0.2% glucose was added to the cultures, phase contrast images were taken 3 hours after induction. (B) Microscopy images were segmented with Omnipose and analyzed for area, length, and width with MicrobeJ. Statistical significance was assessed via Welch's t-test. ****, P < 0.0001.

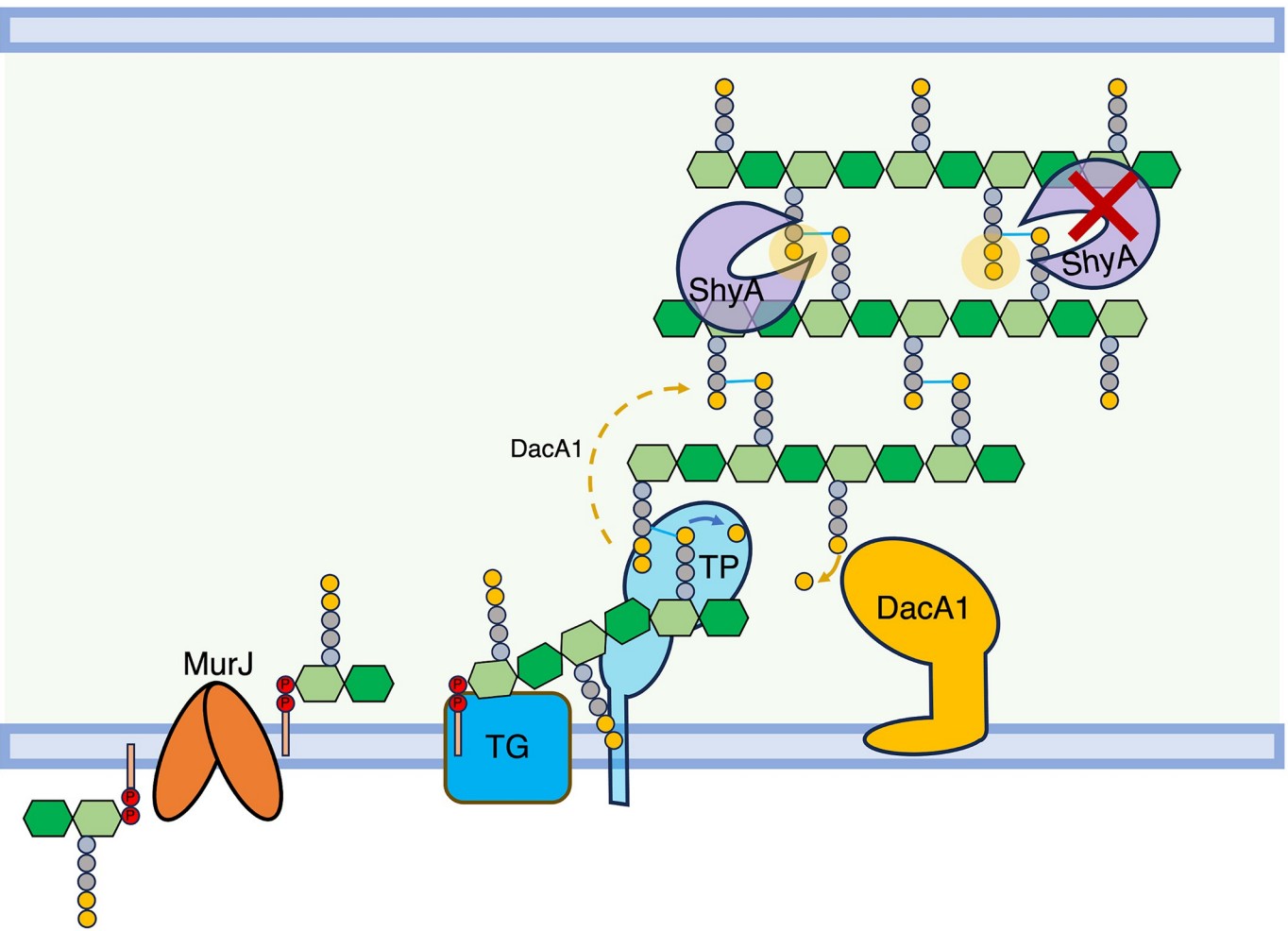

**Fig 8. Model of DacA1's role in balancing PG synthesis and degradation.** DacA1 removes the 5th D-ala residue of the PG side stem and determines which side stem can be used as donor for transpeptidation. By removing the 5th D-ala residue, DacA1 also allows ShyA to cleave crosslinks between strands by producing its preferred substrate, tetrapeptide-containing side stems. In the absence of DacA1, there is an increased amount of pentapeptides in the PG sacculus, hindering ShyA's ability to cleave crosslinks. This reduction in endopeptidase activity disrupts the balance between PG synthesis and degradation.

an increase in pentapeptide side stems, which can serve as both donors and acceptors in trans-peptidation (while the majority of PG crosslinks consist of tetrapeptide and can only serve as acceptor). However, pentapeptide *per se* does not seem to be detrimental—the *V. cholerae* Δ*dacA1* mutant exhibits normal growth and morphology in salt-free LB, despite increased pentapeptide content, suggesting the presence of alleviating factors, and perhaps suggesting another role for carboxypeptidases [27]. Our data here suggest that in addition to promoting incorrect crosslinking, pentapeptide might impede EP-mediated PG cleavage (by increasing the amount of the pentapeptide substrate that is disfavoured by ShyA [7]) (Fig 8), resulting in reduced insertion of nascent PG. Since Δ*dacA1* phenotypes do not mimic simple PG synthase inhibition, however, it is likely that the lack of cleavage has more complex architectural conse-quences. For example, we and others have previously shown that at least in *V. cholerae* and *B. subtilis* (unlike *E. coli* [8]), EP activity is not absolutely required for PG synthesis, but rather for the directional post-synthesis insertion of PG during cell elongation. Though highly specu-lative, this may point towards Höltje's "make before break" model [31], where PG is

constructed as a second layer beneath the first, and then inserted into the structural sacculus by the action of EPs. If this scenario is correct, locally reduced EP activity might create expansive stretches of PG with pentapeptide-rich second layers, locally restricting cell elongation, which could result in the aberrant shapes that are typically associated with lack of carboxypeptidases–like an air balloon being inflated with patches of adhesive tape on its side. This model is consistent with studies done in *E. coli*, where uncharacterized mutations associated with the endopeptidases AmpH and MepA suppress the severe morphological defects in strains missing 3 or 4 PBPs (including DacA)[32]. This model is also consistent with our observations of Δ*dacA1* morphology: An increase in cell width can be attributed to reduced PG insertion activity of the Rod system (SEDS/bPBP2) [33,34].

Our data also reveal a potential reason for the salt sensitivity of the Δ*dacA1* mutant in *V. cholerae*. In contrast to this phenotype, cell wall mutants are often sensitive to low salt conditions due to their inability to withstand their internal osmotic pressure [35]. Our data suggest that surplus PG synthesis is partially responsible for the *dacA1* defects, and perhaps absence of NaCl reduces PG synthesis. *V. cholerae* can generate a sodium motive force (SMF) via its Na (+)-NQR enzyme [36]. Recent work has proposed that SMF is important for cell wall precursor recycling as it can power the C55-P translocase encoded in gene Δ*vca0040* [25], and deletion of *vca0040* indeed reduced salt-sensitivity of the *dacA1* mutant, albeit very modestly. It is formally possible that other cell wall functions across the inner membrane, such as Lipid II flipping by MurJ may also rely on the SMF, boosting PG synthesis in the presence of surplus $Na^+$. MurJ is known to require membrane potential, but not necessarily an H+ gradient to function. The specific ions involved in lipid II transport by MurJ depend on the organism; for instance, *Thermosipho africanus* requires chloride ions for MurJ function, while MurJ from *E. coli* uses the proton motive force [37]. Interestingly, a recent report demonstrated sensitivity to very high salt (750 mM) in the *dacA* mutant in *E. coli* as well [38]. Since we do not expect a sodium motive force to promote cell wall synthesis in *E. coli*, this sensitivity may rather reflect non-specific physiology, e.g., disruption of protein-protein interactions that is detrimental only in the absence of PBP5. In addition to a role for $Na^+$ in boosting PG synthesis, it is also possible that salt concentrations directly affect ShyA activity, by altering the hydrophobic interactions involved in its structural switching, leading to decreased activity in high salt conditions and increased activity in salt-free conditions.

Broadly, or data imply that the balance between PG synthesis and degradation does not necessarily have to be maintained by direct protein-protein interactions (tight complexes between hydrolases and synthases), since the same end-result can be achieved by upregulating degradation enzymes in diverse ways and downregulating synthesis in diverse ways. A physical uncoupling between PG synthesis and degradation may explain the high redundancy in synthases and hydrolases, as this allows for the maintenance of this balance under varying environmental conditions without relying on strong and specific protein complexes.

## Methods

### Bacterial strains, media, and growth conditions

All *V. cholerae* strains used in this study are derivatives of El Tor strains N16961 and C6706 (S4 Table). *V. cholerae* cultures were grown by shaking (200rpm) at 30˚C in LB broth (tryptone 10g/L, yeast extract 5g/L, sodium chloride 10g/L) or salt-free LB (tryptone 10g/L, yeast extract 5g/L) unless otherwise indicated. 200μg/mL of streptomycin was also added to the cultures (the *V. cholerae* strains used are streptomycin resistant). When appropriate, the overnight cultures were prepared with 0.2% glucose to prevent the leaky expression from the $P_{IPTG}$ promoter. For growth curve experiments, overnight cultures were diluted 100-fold into 1 mL of

the appropriate growth medium containing streptomycin and incubated at 37°C with shaking in 100-well honeycomb wells in a Bioscreen plate reader (Growth Curves America) which recorded optical density at 600 nm (OD600) in 10-minute intervals. *E. coli* cultures were grown by shaking (200rpm) at 37°C in LB broth unless otherwise stated. When appropriate, antibiotics were used in the following concentrations: carbenicillin (100 μg/mL), kanamycin (50 μg/mL) and chloramphenicol (20 μg/mL). Genes under $P_{IPTG}$ regulation were induced with 1mM IPTG while genes under $P_{BAD}$ regulation were induced with 0.8% arabinose, both systems were repressed with 0.2% glucose in both *V. cholerae* and *E. coli*.

## Cloning, vectors, and strain construction

A summary of the plasmids and primers used in this study can be found in S6 and S7 Tables. *E. coli* DH5α *λ pir* was used for general cloning, *E. coli* SM10 and MFD *λ pir* were used for conjugation into *V. cholerae* (S5 Table). Plasmids were built using isothermal assembly [39] and verified by Sanger Sequencing.

Chromosomal in-frame deletions were generated using the allelic exchange vector pTOX5 cmR/msqR[40]. Regions flanking the gene to be deleted (500-700bp) were amplified from *V. cholerae* N16961 genomic DNA, cloned into the suicide vector pTOX5 and transformed into *E. coli* DH5α. Chloramphenicol resistant colonies were verified by colony PCR using primers MA83 and MA84 and verified via Sanger sequencing. Correct constructs were selected and transformed into *E. coli* SM10 (Donor) and Conjugated into *V. cholerae* (recipient) by spotting 15μL of the donor and 15μL of the recipient onto a LB plate with 1% glucose and incubating for 4h at 37°C. Transconjugants were selected on LB plates with chloramphenicol (20 μg/mL), streptomycin (200 μg/mL) and 1% glucose after incubation at 30°C overnight. Chloramphenicol- and streptomycin- resistant colonies were grown without selective pressure in microcentrifuge tubes with 1mL LB broth with 1% glucose and incubated for 3h at 37°C. 2–3 candidates were then streaked on M9 minimal medium supplemented with 0.2% (wt/vol) casamino acids, 0.5 mM MgSO4, 0.1 mM CaCl2, 25 μM iron chloride in 50 μM citric acid and 2% rhamnose, and grown at 30°C. Single colonies were further purified by streaking onto M9-rhamnose plates and deletions were verified by PCR using flanking and internal primers for each gene and validated by whole genome sequencing. Given *ΔdacA1* salt-sensitivity, all cloning steps with LB media were replaced with salt-free LB.

For complementation of deletion strains, genes were amplified from *V. cholerae* N16961 with primers designed to introduce a strong ribosome-binding site (RBS) (AGGAGG) and cloned into pTD101 a pJL1 [41] derivative, which contains *lacI^q*, a multiple cloning site downstream of an IPTG (isopropyl-β-D-thiogalactopyranoside)-inducible promoter ($P_{IPTG}$) and allows chromosomal insertions of the gene of interest into the native *lacZ locus* in *V. cholerae*. Successful insertions were verified by PCR using lacZ flanking primers (MA106-107).

To replace the chromosomal copies of *murA* and *murD* with the mutants isolated from the suppressor screen, we used the allelic exchange vector pTOX5 as described above, however, instead of just using the *murA* and *murD* flanking regions, we used the gDNA from the suppressor mutants and used primers that annealed 500bp upstream and 500bp downstream to amplify the mutant gene with its corresponding flanking regions for insertion into the chromosomal WT locus. Successful integration was validated by PCR amplification and Sanger sequencing.

ShyA^R115W mutant constructs were generated by amplifying genomic DNA from the suppressor strain and cloned into pTD101 and pHLmob100. Constructs were verified using Sanger sequencing.

## TnSeq

TnSeq was conducted as described before [42,43]; Briefly, cultures of strains MA1 and MA73 were mated with donor strain *E.coli* MFD λ pir, which contains the pSC189 suicide plasmid encoding the mariner transposon. Approximately 200,000 colonies/replicate were recovered from plates containing Kan50 to select for transposon insertions and 1mM IPTG to allow the expression of ShyA$^{L109K}$. Libraries were then resuspended in 30 mL of LB broth, and 1/5 of the culture was used for genomic DNA (gDNA) extraction and the rest was frozen in 30% glycerol at -80°C. Samples were prepared for sequencing as follows. The extracted gDNA was sheared by sonication (9 seconds, 30% amplitude), followed by blunting (Blunting Enzyme Mix, NEB), A-tailing, ligation of specific adaptors and PCR amplification of the transposon-DNA junctions using transposon- and adaptor- specific primers. Libraries were sequenced using Illumina MiSeq as described previously [44]. To determine gene essentiality, data analysis was performed using the Matlab-based pipeline ARTIST [43]. Genetic regions predicted to be conditionally essential/enriched were inspected using the genome browser Artemis [45] and insertion plots were generated using the tidyverse package in R.

## PG purification and Remazol Brilliant Blue assay of hydrolase activity

Peptidoglycan sacculi were purified as described previously [46]. Briefly, 1 L of WT and *ΔdacA1* cultures were harvested and resuspended in boiled 5% SDS for 30 minutes. The cell lysate was pelleted by ultracentrifugation at 130,000g for 60 minutes at room temperature and resuspended in 30mL of MilliQ water at 50°C, the pellets were washed 4 times before being resuspended in 100mM Tris HCl pH 7.5 with 100 ug of trypsin and CaCl2 to a final concentration of 10 mM and incubated at 37°C overnight. On the next day, samples were boiled for 15 min to inactivate the Trypsin, washed twice with 30 mL of warm MilliQ water and centrifuged at 130,000g for 30 minutes at 20°C. The pellet was then resuspended in 500uL of MilliQ water and stored at -80°C.

Purified sacculi were stained with Remazol Brilliant Blue (RBB) as described previously [47]. In a 15mL conical tube, 1mL of sacculi was mixed with 0.4 mL of 0.2M RBB (Sigma, R8001), 0.3 mL of 5M NaOH and 4.3 mL of MilliQ water and incubated at 37°C overnight on a rocker table. To neutralize the pH and wash away the residual dye, 0.4 mL of 5M HCl was added alongside 0.75 mL of 10X PBS, samples were then ultracentrifuged at 400,000g for 20 min at 22°C and resuspended in MilliQ water, the samples were washed multiple times until the supernatant ran clear. To eliminate possible contamination with lysozyme, the stained sacculi were incubated at 65°C for 3 hours and stored in 10%glycerol at -20°C.

Endopeptidase activity reactions were prepared with 200μL of Pulldown buffer[6] (20 mM tris/maleate (pH 6.8), 30 mM Nacl, 10 mM MgCl, 1 mM DTT, 0.1% triton X-100), 50uL of RBB-stained sacculi and 10 μg of lysozyme purified ShyA and ShyA$^{L109K}$, and incubated at 37°C for 24 h. The reactions were stopped by adding 1% SDS and washed by ultracentrifugation at 80,000g for 15 min. Supernatants were removed and A$_{585}$ measured in 96-well plates using a microplate reader.

## Peptidoglycan profiling

UPLC analyses were performed on a Waters UPLC system equipped with a Kinetex C18 Column, 100 Å, 1.7 μm, 2.1 mm × 150 mm (Waters Corporation) and quantified by A$_{204}$ nm. Muropeptides were separated using a linear gradient from buffer 0.1% formic acid in water (A) to buffer 0.1% formic acid in acetonitrile (B). Identification of individual peaks was assigned by the mass of the muropeptide quantified by a Xevo G2/XS Q-TOF MS (quadrupole time-of-flight mass spectrometry) instrument coupled to the system. Instrument was operated

in DDA mode and positive ionization using the following parameters: capillary voltage at 3.0 kV, source temperature to 120°C, desolvation temperature to 350°C, sample cone voltage at 40 V, cone gas flow 100 L/h, desolvation gas flow 500 L/h, and collision energy (CE) ramp; low CE 20–40 eV and high CE 30–60 eV. Mass spectra were acquired every 0.05 s/scan in the range of 100–2,000 m/z. MassLynx software package was used for data acquisition and processing. The relative amount of each muropeptide was calculated by the ratio of each peak area of the muropeptide to the total area of the $A_{204}$ nm chromatogram. The abundance of peptidoglycan (total PG) was calculated relative to the total area of the chromatogram to the $A_{600}$. The cross-linking values were calculated as described previously [48].

## Microscopy and image analysis

Cells were grown as previously described and imaged under phase contrast on an agarose pad (0.8% agarose in LB or salt-free LB (SF)) using a Leica DMi8 inverted microscope. Images were segmented using Omnipose [49] using the bact_phase_omni model, masks were exported as.png and imported into MicrobeJ [50] for image analysis and quantification using the default parameters. Significant differences were determined using Welch's two-sample t-test. Plots were made using the R packages ggplot2 and tidyverse.

## Spontaneous suppressor identification

*ΔdacA1* cells were grown overnight in SF medium, 1mL of overnight culture was spread on a BioAssay square dish (Thermo Scientific Nunc) containing 200 mL of LB medium with streptomycin 200 μg/mL. This experiment was performed 3 independent times, for each instance, 16 colonies were selected from the plate, to validate the stability of the mutations each colony was grown overnight in salt-free LB and then in LB agar. Genomic DNA from each selected colony was extracted and samples were sent to SeqCenter (Pittsburg, PA) for whole genome sequencing and variant calling analysis.

## Growth curves

Saturated overnight cultures were diluted 1:10,000 into 200 μL of LB medium and incubated in a Bioscreen growth plate reader (Growth Curves America) at 37°C with random shaking at maximum amplitude, and $OD_{600}$ recorded at 5 min intervals for 24h.

## Western Blot analysis

Expression of the different ShyA mutants (L109K and R115W) was assessed via Western Blot. Overnight cultures of WT (LB medium) and *ΔdacA1* (SF medium) strains harboring an IPTG-Inducible of ShyA, ShyA$^{L109K}$ and ShyA$^{R115W}$ as well as the empty vector control were subcultured 1:100 into LB medium and grown at 37°C until they reached an $OD_{600}$ between 0.5–0.6. After the cultures reached the desired $OD_{600}$ they were induced with 1mM IPTG for two hours. Cultures were normalized by $OD_{600}$ and cells were harvested by centrifugation (9500 × g, 15 min) at 4°C and resuspended in 150μL of dH$_2$O and 150μL of 2X lysis buffer (0.1M Tris-HCL pH 8.0, 0.5 M NaCl and 0.2% SDS). Resuspended cells were incubated at 95°C for 3 min, then sonicated 4 × 5 s at 20% amplitude. Western blot against ShyA was performed as described previously [12]. Briefly, ShyA was detected by using an anti-ShyA polyclonal antibody (1:5,000, produced by Pocono Rabbit Farm & Laboratory, PA) and an anti-rabbit secondary antibody (1:15,000, Li-Cor cat# 926–32211) and scanned on an Odyssey CLx imaging device (LI-COR Biosciences), and visualized using Image Studio Lite (Li-Cor) software. RpoA was used as loading control, the same membranes were incubated with

monoclonal anti-αRpoA antibody (1:15,000, BioLegend cat# 663104) and an anti-mouse secondary antibodies (1:15,000, Li-Cor cat# 926–32210), images were obtained as previously described.

## Protein purification

MurA, MurA$^{P122S}$, MurA$^{L35F}$, MurC and MurC$^{A132T}$ genes were amplified from N16961 gDNA and the gDNA from the mutants obtained from the suppressor screen, and cloned into pET28a downstream of 6xHis-SUMO Tag. Plasmids were verified by PCR and Whole Plasmid Sequencing through Plasmidsaurus (Eugene, OR). *E. coli* BL21 (DE3) (Novagen) was transformed with the resulting recombinant plasmids (pET28a-MurA, pET28-MurA$^{P122S}$, pET28a-MurA$^{L35F}$, pET28a-MurC and pET28a-MurC$^{A132T}$). Overnight cultures (10 mL) were used to inoculate 1L of LB with kanamycin (50 μg/mL) and incubated at 37˚C with vigorous shaking (220 RPM) until they reached an OD$_{600}$ between 0.6 and 0.8. Cultures were induced with 1mM IPTG at 18˚C and 180 RPM overnight. Harvested cells were pelleted and resuspended in 30 mL of cold purification buffer (20mM Tris pH 7.5, 150 mM NaCl, 1mM PMSF), and lysed by sonication. Lysates were cleared by centrifugation at 31,000g for 4 minutes at 4˚C, and loaded onto a HisPur cobalt column (Thermo Scientific; Catalog No. 89964) and washed multiple times with purification buffer until protein was undetectable by the Bradford reagent. The bead slurry was then transferred to a 5 mL microtube with 60 μL of ULP1 Sumo protease and digested overnight at 4˚C rotating. Protein was eluted the next day with 20 mL of purification buffer. Samples were analysed by SDS-page with Coomassie blue stain and then Concentrated with a 30KD Amicon concentrator (Millipore) to 5 mL. Concentrated samples were then passed through a HiLoad 16/600 Superdex 75pg gravity column using and ÄKTA pure chromatography system (Cytiva). ShyA and ShyA$^{L109K}$, were purified as previously described [51].

## MurA enzymatic activity

The enzymatic activity of MurA, MurA$^{P122S}$ and MurA$^{L35F}$, was measured using a bacterial MurA Assay (Profoldin, Hudson MA) based on the spectrophotometric measurement of inorganic phosphate released from the enzymatic reaction catalysed by MurA. This reaction transfers enolpyruvate from phosphoenolpyruvate (PEP) to uridine diphospho-N-acetylglucosamine (UNAG), which results in the generation of enolpyruvyl-UDPN-acetylglucosamine (EP-UNAG) and inorganic phosphate.

For this experiment MurA$^{E.coli}$ provided by the manufacturer was used as a control, and 1μL of 50mg/mL fosfomycin was added to the WT versions of MurA to validate the ability of the test to detect reduced activity of MurA. The reactions were performed by triplicate following the manufacturer's protocol. Briefly 5000mM of each protein was resuspended in the reaction buffer (for 1 reaction: 6.6 μL of 10X buffer and 52.2 μL of H$_2$O) and the enzyme substrate was prepared (for 1 reaction: 0.66 μL of 100X PEP, 0.66 μL of 100X UGN and 5.28 μL of H$_2$O), the reactions were prepared by adding 54 μL of resuspended MurA and 6 μL of the enzyme substrate to a 96-well plates, samples were incubated at 37 ˚C for 60 min. For inorganic phosphate detection, 90μL of Dye MPA3000 was added and samples were incubated for 5 mins before measuring light absorbance at 650nm in a 96-well plate reader.

## MurC enzymatic activity

The enzymatic activity of MurC and MurC$^{A132T}$, was measured using a bacterial MurC Assay (Profoldin, Hudson MA) based on the spectrophotometric measurement of inorganic phosphate released from the enzymatic reaction catalysed by MurC. This enzyme catalyses the

addition of L-alanine to UDP-MurNAc generating UDP-MurNAc-L-Ala, with a reaction coupled to ATP hydrolysis which results in the formation of ADP and inorganic phosphate.

For this experiment $MurC^{E.coli}$ was used as a positive control for MurC activity and purification buffer was used as a negative control. The reactions were performed by triplicate following the manufacturer's protocol. Briefly 5000mM of each protein was resuspended in the reaction buffer (for 1 reaction: 6.6 μL of 10X buffer and 52.2 μL of $H_2O$) and the enzyme substrate was prepared (for 1 reaction: 0.66 μL of 100X UMA, 0.66 μL of 100X D-Glu, 0.66 μL of 100X ATP and 4.62 μL of $H_2O$), the reactions were prepared by adding 54 μL of resuspended MurC and 6 μL of the enzyme substrate to a 96-well plates, samples were incubated at 37 ˚C for 60 min. For inorganic phosphate detection, 90μL of Dye MPA3000 was added and samples were incubated for 5 mins before measuring light absorbance at 650nm in a 96-well plate reader.

## Supporting information

**S1 Fig. Overexpression of ShyA$^{L109K}$ reduces the morphological defects in *ΔdacA1* cells.**
(A) Phase contrast images were segmented using omnipose and area, width and length were calculated using MicrobeJ. Statistical significance was assessed via Welch's t-test. ****, P < 0.0001 (B) Overnight cultures were grown on LB (WT background) and salt-free LB (*ΔdacA1* background) and then spot-titered on either LB or salt-free LB with 1mM IPTG. (C) Length and width of WT and *ΔdacA1* cells grown in salt-free LB were calculated using MicrobeJ. Statistical significance was assessed via Mann Whitney U test. (P < 0.05 was interpreted as statistically significant) (D) Western Blot analysis of ShyA levels post IPTG induction. Overnight cultures were sub-cultured 1:100 in LB and incubated at 37˚C until they reached exponential phase, followed by addition of inducer (1mM IPTG). Protein samples were taken after 2h of induction.
(TIF)

**S2 Fig. Overexpression of ShyB, ShyA and ShyA$^{R115W}$ significantly reduces the morphological defects of ΔdacA1 cells in LB medium.** Cells were grown as described in Fig 4 legend. The microscopy images were segmented using Omnipose and then analyzed for their (A) area, (B) length, and (C) width using MicrobeJ. Statistical significance was determined using Welch's t-test. The significance level was denoted as ****, indicating a p-value of less than 0.0001. (D) Overnight cultures were grown on LB (WT background) and salt-free LB (*ΔdacA1* background) and then spot-titered on LB medium containing either 0.2% glucose or 1mM IPTG.
(TIF)

**S3 Fig. Overexpression of other D,D-endopeptidases encoded in *V. cholerae* genome partially rescue the *ΔdacA1* morphological defects.** Overnight cultures were grown on LB (WT background) and salt-free LB (*ΔdacA1* background), then sub-cultured 1:100 in LB medium for 2 hours after which 1mM IPTG was added to the cultures, which where then incubated for 1h. (A) Microscopy images after induction. (Scale bar: 10 μm). (B) Microscopy images were segmented using Omnipose and analyzed using MicrobeJ for area, length and width. Statistical significance was determined via one-way ANOVA. The significance level was denoted as ns, not significant; *, P < 0.05; **, P < 0.01; ***, P < 0.001; ****, P < 0.0001.
(TIF)

**S4 Fig. Overexpression of ShyA$^{R115W}$ is toxic in *E. coli*, it induces spheroplast formation and lysis consistent with excessive endopeptidase activity.** (A) Two strains of E. coli, CS109 and Δ4 harboring the indicated vectors (pH100mob empty, pHL100mob ShyA or pHL100mob

ShyA$^{R115W}$), were incubated overnight at 37˚C in LB medium with 0.2% glucose. Afterward, they were sub-cultured at a 1:100 dilution in LB and incubated for 2 hours at 37˚C. The cultures were then sub-cultured at a 1:100 and incubated for an additional 2 hours, after which either 1mM IPTG or 0.2% glucose was added. 1 hour after induction, cells were plated on LB with 1mM IPTG or 0.2% glucose and (B) phase contrast images were captured (Scale bar = 10μm).
(TIF)

**S5 Fig. Mutations in MurA, MurC and MurD map to regions important for protein function.** Protein structures were predicted using Alphafold2, domains predicted using interpro and mutated residues were identified using ChimeraX. (A) The MurA structure depicts the two globular domains and the catalytic cysteine as well as residues P122 and L35 (light blue). Mur ligases (B) MurC and (C) MurD contain 3 functional domains, Domain 1 binds the nucleotide substrate (yellow), Domain 2 binds ATP and contains the ATP-binding consensus sequence (GKT) (red) and Domain 3 binds the aminoacid substrate (green). The MurC dimer was predicted using alphafold multimer. Residues MurC (A132) and MurD (D447) are colored in light blue.
(TIF)

**S6 Fig. Mutations in *mur* genes exacerbate the *ΔdacA1* morphological defects.** Cells were grown as described in Fig 5B. Images were segmented using Omnipose and then analyzed for their area, length, and width using MicrobeJ. Statistical significance was determined using Welch's t-test. The significance level was denoted as ns, not significant; *, P < 0.05; **, P < 0.01; ***, P < 0.001; ****, P < 0.0001.
(TIF)

**S7 Fig. Phenotypic characterization of the mur mutants.** (A) Overnight cultures were diluted 1:10,000 and grown on LB broth for 20 hours, OD600 measurements were taken every 2 minutes. (B) Fosfomycin sensitivity was measured in the different *murA* mutants. Cultures of the indicated strains were plated evenly ($\sim 10^8$ colony forming units [cfu]) on LB agar. A filter disk containing 10 μL of fosfomycin (50mg/mL) was placed in the center of the plate followed by incubation at 30˚C for 24h. Bars indicate zone of inhibition (ZOI) in mm. Statistical significance was calculated with a one-way ANOVA. Significance level was denoted as ns, not significant; *, P < 0.05; **, P < 0.01; ***, P < 0.001; ****, P < 0.0001.
(TIF)

**S8 Fig. Mutation L35F in *murA* results in a significant decrease of PG amount in the *ΔdacA1* background.** Relative PG amount was calculated by measuring the total area of the chromatogram and normalized to the OD at the time of harvesting. The relative percent was calculated by dividing the normalized total area by the average of the control sample and multiplying it by 100. Statistical significance was calculated using an unpaired *t-test* analysis (*P-value < 0.05, **P-value < 0.01, ***P-value < 0.001).
(TIF)

**S9 Fig. Fosfomycin treatment in *ΔdacA1* cells partially restores growth in LB.** Overnight cultures were plated on LB plates with increasing concentrations of fosfomycin (5μg/mL, 10μg/mL, 15μg/mL, 30μg/mL) and incubated overnight at 30˚C.
(TIF)

**S10 Fig. Independent treatment with antibiotics that target the rod complex, aPBPs or the divisome does not rescue the ΔdacA1 phenotype in LB.** Overnight cultures were spot-plated on LB plates with different concentrations of MP265 (100 μg/mL, 5x MIC and 200 μg/mL, 10x

MIC), A22 (1 μg/mL, 0.5x MIC, 5 μg/mL, 2.5x MIC and 10 μg/mL, 5x MIC), Mecillinam (0.1 μg/mL, 0.2x MIC, 1 μg/mL, 2x MIC and 5 μg/mL 10x MIC), Moenomycin (0.1 μg/mL, 0.08x MIC and 1 μg/mL, 0.8x MIC) or aztreonam (0.1 μg/mL, 0.02x MIC) and incubated overnight at 37°C. A and B are Independent biological replicates.
(TIF)

**S11 Fig. Combination treatment with inhibitors of the Rod system and aPBPs partially rescues *dacA1* mutant growth.** Overnight cultures were spot-plated on LB plates with different combinations of sub-MIC concentrations of A22 (1 μg/mL, 0.5x MIC and 5 μg/mL, 2.5x MIC) and moenomycin (0.1 μg/mL, 0.08x MIC, 0.5μg/mL, 0.4x MIC and 1 μg/mL, 0.8x MIC) and incubated overnight at 37°C. A, B and C are independent biological replicates.
(TIF)

**S12 Fig. Deletion of VCA0040 partially rescues *ΔdacA1* growth in LB.** *vca0040* under control of an IPTG-inducible promoter was inserted chromosomally into the *lacZ* locus in the *ΔdacA1ΔVCA0040* background, overnight cultures were spot-titered on LB medium with 0.2% glucose or 1mM IPTG and incubated at 30°C overnight. (A) Replicate 2 and Replicate 3 (empty pTD101 vector integration as negative control). (B) Overnight cultures were diluted 1:10,000 and grown on LB broth for 20, OD600 measurements were taken every 2 minutes. (C) Microscopy images taken after 2 subcultures in LB medium. (Scale bar: 10μm) (D) Cell size calculations done with MicrobeJ using segmentation masks generated with Omnipose. Statistical significance was determined using ANOVA, **** corresponds to a P-value of <0.0001.
(TIF)

**S13 Fig. Overexpression of MurA in *ΔdacA1* cells increases salt sensitivity.** Overnight cultures of WT, *ΔdacA1* and both backgrounds harbouring an IPTG inducible copy of *murA* were plated on salt-free LB with increasing concentrations of NaCl (120mM, 140mM and 160mM) and either 0.2% glucose or 1mM IPTG, and incubated at 30°C overnight. (A) Replicate 2 (B) Replicate 3 with empty pTD101 vector integration as negative control and 3 different clones of *ΔdacA1* P$_{IPTG}$-*murA*.
(TIF)

**S14 Fig. Expression of MepMΔdom1 and ShyB rescues growth and morphological defects in the Δ4 strain.** (A) Two strains of E. coli, CS109 and Δ4 harboring the indicated vectors (pBADmob-empty, pBADmob MepM, pBADmob MepMΔdom1 or pBADmob ShyB), were incubated overnight at 37°C in LB medium with 0.2% glucose. Afterward, they were sub-cultured at a 1:100 dilution in LB and incubated for 2 hours at 37°C. The cultures were then sub-cultured at a 1:100 dilution and incubated for an additional 2 hours, after which either 0.8% arabinose or 0.2% glucose was added. 1 hour after induction, cells were plated on LB with 0.8% arabinose or 0.2% glucose and (B) phase contrast images were captured (Scale bar = 10μm). (C) Microscopy images were segmented using Omnipose and analyzed using MicrobeJ for area, length and width. Statistical significance was determined via one-way ANOVA. The significance level was denoted as ns, not significant; *, P < 0.05; **, P < 0.01; ***, P < 0.001; ****, P < 0.0001.
(TIF)

**S1 Table. Processed TnSeq Data.**
(XLSX)

**S2 Table. Suppressor screen mutation summary (*ΔdacA1* on LB).**
(DOC)

**S3 Table. PG analysis.**
(DOC)

**S4 Table.** *V. cholerae* **strains used in this study.**
(DOC)

**S5 Table.** *E. coli* **strains used in this study.**
(DOC)

**S6 Table. Plasmids used in this study.**
(DOC)

**S7 Table. Oligonucleotides used in this study.**
(DOC)

## Acknowledgments

We thank Matt Jorgenson and Kevin Young (University of Arkansas) for the gift of CS109 and the Δ4 derivative, and Laura Álvarez for her invaluable help and support for the peptidoglycan profiling.

## Author Contributions

**Conceptualization:** Manuela Alvarado Obando, Tobias Dörr.

**Data curation:** Manuela Alvarado Obando, Diego Rey-Varela.

**Formal analysis:** Manuela Alvarado Obando, Diego Rey-Varela, Tobias Dörr.

**Funding acquisition:** Felipe Cava, Tobias Dörr.

**Investigation:** Manuela Alvarado Obando, Diego Rey-Varela, Tobias Dörr.

**Methodology:** Manuela Alvarado Obando, Diego Rey-Varela, Tobias Dörr.

**Project administration:** Tobias Dörr.

**Resources:** Felipe Cava, Tobias Dörr.

**Supervision:** Felipe Cava, Tobias Dörr.

**Validation:** Manuela Alvarado Obando, Diego Rey-Varela, Tobias Dörr.

**Visualization:** Manuela Alvarado Obando.

**Writing – original draft:** Manuela Alvarado Obando, Tobias Dörr.

**Writing – review & editing:** Manuela Alvarado Obando, Diego Rey-Varela, Felipe Cava, Tobias Dörr.

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
