## [Decision Letter · Decision Letter 0]

5 Nov 2023

Dear Dr Dörr,

Thank you very much for submitting your Research Article entitled 'Genetic interaction mapping reveals functional relationships between peptidoglycan endopeptidases and carboxypeptidases.' to PLOS Genetics.

The manuscript was fully evaluated at the editorial level and by independent peer reviewers. The reviewers appreciated the attention to an important problem, but raised some substantial concerns about the current manuscript. Based on the reviews, we will not be able to accept this version of the manuscript, but we would be willing to review a much-revised version. We cannot, of course, promise publication at that time.

If you decide to revise the manuscript for further consideration at PLOS Genetics, please aim to resubmit within the next 60 days, unless it will take extra time to address the concerns of the reviewers, in which case we would appreciate an expected resubmission date by email to plosgenetics@plos.org.

Please do not hesitate to contact us if you have any concerns or questions.

Yours sincerely,

Jan-Willem Veening, Ph.D.

Academic Editor

PLOS Genetics

Lotte Søgaard-Andersen

Section Editor

PLOS Genetics

Reviewer's Responses to Questions

**Comments to the Authors:**

Reviewer #1: This paper convincingly shows a close relation between CPase and EPase activity. Loss of CPase activity results in increased PG with pentapeptide stems, which are less efficiently cleaved by EPases – in order for the cell to survive, EPase activity needs to be increased, or PG synthesis needs to be decreased, which presumably results in better control of the number of crosslinks in the wall.

DacA1 is essential when cells are grown at high salt. The Tn Seq screen identifies DacA1 deletion as a way to rescue lethal overexpression of a ShyA hyperactive mutant (line 132-133). A screen for DacA1 suppressors identifies another ShyA activating mutant (lines 189-199). ShyA activation is mediated by hydrophobic interactions between protein domains. It seems plausible that ShyA activity itself is salt-sensitive – and that it is more active at low salt. Have the authors tested this? For example, the PG degradation assays (Fig 2) could be performed in high salt (current buffer is 30 mM which would be a lot lower than in LB). Also, if L109K is a hyper active mutant, why is there no discernable difference in the in vitro activity of ShyA and ShyAL109K on wildtype sacculi? Is the assay saturated? What is the activity in Fig. 2C relative to – none of the bars matches ‘1’ so this is hard to see?

Also – the rescue of the dacA1 phenotype by ShyAR115W and ShyB overexpression – the results in table S3 reveal that these two constructs are capable of a significant reduction of the overall number of crosslinks – what one would expected with an active EPase. What happens to the number of crosslinks in the wildtype when grown on high salt (I assume all results in TS3 are low salt since the dDacA1 cannot grow on high salt)?

In short – the authors link salt sensitivity to vca0040 (c55P translocase) but indicate that this cannot be the sole explanation, and salt sensitivity is conserved in E.coli where the SMF is not expected to play a role. I would expect a more thorough investigation of the effects of salt on the activities of both DacA1 and ShyA (mutants). If this already has been done in previous studies this should be stated more explicitly.

Minor comments

- The authors should specify growth conditions for experiments in all figure or table legends given the importance of the presence/absence of salts for the phenotypes.

- The introduction is phrased very ‘general’ – as if the statements made are valid for all bacteria. This is not the case, the authors should be more specific. Examples:

L 47-50: description of PG, stem peptide composition, is correct for typical Gram-negative PG but reads as if valid for all bacteria.

L 66: pentapeptide is not “exclusively” associated with newly synthesized PG – eg Neisseria meningitidis has relatively high levels of pentapeptide in its PG (https://doi.org/10.1371/journal.pgen.1005338) and B. subtilis had pentapeptides enriched at division sites (https://doi.org/10.1111/mmi.13629).

- L 71 DacA not DacA1.

- L112 if ShyAL109K is already hyperactive, why does it need to be overexpressed as well?

- L 114 – ref 12 is preprint of ref 11-

- L 121 and further – the use of the phrase ‘the screen was answered’ – I find this an odd turn of phrase and don’t recall reading it elsewhere in papers using TnSeq. Matter of taste but I would rephrase.

- L 213-4 D44 and D45 should be explained.

- L 224 – “hypomorph” – I dno’t know what is meant by this expression.

- L349-50 “tetrapeptide and can only serve as donor” – shouldn’t this be “acceptor”?

Reviewer #2: review is attached as a separate document

Reviewer #3: In this study, Alvarado-Obando et al. revealed a functional interaction between endopeptidases (EPs) and the main carboxypeptidase of Vibrio cholerae, DacA-1. They began by using a strain overexpressing a hyperactive mutant of the EP ShyA (ShyAL109K) and identified through Tn-seq the genes that are essential for tolerating ShyAL109K toxic activity, as well as those whose inactivation helps to tolerate it. The genes that became essential were found to be associated with peptidoglycan synthesis (pbp1a, csiV), recycling (ampG), and remodeling (mltB). Intriguingly, a gene typically considered essential under the conditions of the screen, dacA-1, had an unusually high number of transposon insertions.

A prior study by the same group demonstrated that ShyA primarily targets the peptidoglycan (PG) matrix with tetrapeptides as stem peptides (referred to as "mature" PG). In this study, they provide compelling genetic and biochemical data identifying DacA-1 as responsible for modifying pentapeptides to become the preferred substrate for ShyA hydrolysis. In absence of DacA-1, the PG matrix contains mostly pentapeptides and is less subject to ShyAL109K hydrolysis.

Subsequently, they adopted the reverse approach, seeking spontaneous mutations that enabled the growth of the strain deleted for the essential dacA-1 gene in LB medium (which contains salt). As expected from the previous presented data, they identified mutations in pathways leading to upregulation of EPs (ShyA or ShyB) activity. Additionally, they got mutations that reduce synthesis of PG precursor (murA, murC and murD genes).Intriguingly, only the EPs could correct both the growth and the morphology defect of the dacA-1 mutant. The mur gene could only restore formation of colonies on LB medium, but the cell shape was still perturbed. The authors conclude that the ∆dacA-1 strain suffers from an excess PG synthesis that can be partially corrected by reducing the level of PG precursor, by inhibiting PG synthesis with antibiotics or by increasing PG degradation (increase of EPs activity).

The authors then endeavored to uncover the cause of salt sensitivity of the dacA-1 mutant. Recently, the translocase responsible for recycling the lipid carrier involved in PG precursor synthesis (C55-P), known as Vca0040, was identified. This enzyme might be powered by the sodium motive force. The authors hypothesized that in the absence of salt (sodium), the recycling of C55 is likely affected, which in turn reduces the formation of PG precursor, leading to reduced PG synthesis. The authors conducted experiments to confirm that a mutation in vca0040 had a beneficial but minimal effect on the ∆dacA-1 mutant.

This study presents a lot of results. I am impressed by the efforts put into the biochemical experiments. The results are clearly presented and well interpreted. The functional connection between DacA-1 and ShyA is clearly established, however the suppression of DacA-1 essentiality remains a little bit unclear. A lot of suppressor mutations mapped in LPS modification enzymes (WaaL and WbeT) but the authors did not elaborate on them. If these specific mutations will be the object of another study, the authors should mention it.

I recommend publishing this study in PLoS Genetics once the following points, listed below, have been addressed.

Major points:

Lane 130-136: How does the Tn-seq profile look for the other genes encoding enzymes with carboxypeptidase activity (dacA-2, dacB and pbpG)? As they have little influence on the percentage of pentapeptides in the PG matrix in these conditions, they should not help to tolerate ShyAL109K toxicity.

Lane 183-187: Overexpression of ShyB can suppress ∆dacA-1 growth defect. Is ShyB activity affected by the presence of the pentapeptide content in the PG matrix, like in the case of ShyA? V. cholerae genome encode many additional endopeptidases. Can any of them suppress ∆dacA-1 defects?

Lane 195-197: What is the level of the ShyA, ShyAR115W and ShyAL109K proteins. Could the difference in toxicity in the WT and ability to rescue growth and morphology in the ∆dacA-1 depends on the protein level or protein stability?

Lane 232-233: Is the murA::murAL35F strain more sensitive to Fosfomycin? As it has reduced activity (Fig. 5C), but the WT with this mutation seems to be perfectly happy (no shape or plating defects), we might at least expect a higher sensitivity to Fosfomycin.

Lane 236-237: Is there a specific reason why the authors did not reconstitute the murCA132T mutation in the WT and ∆dacA-1 strain? Were there any technical issues?

Lane 297-300: How does the double mutant ∆dacA-1∆vca0040 cells look like? As this mutation should impact PG synthesis by affecting the level of PG precursor, we might expect a phenotype similar to the murA, murC or murD mutations and no improvement of the cell morphology.

Lane 304-305: The cited paper (Hernández et al. , 2020) used a Pbad plasmid to overexpress murA, which is quite different from the single integration used by the authors here. The effect of murA overexpression might be stronger if murA was expressed from a similar replicative plasmid.

Lane 323-325: Have you tried to overexpress shyB or an E. coli endopeptidase to correct the ∆4 defects? In the cited paper, Laubacher et al ., 2013 did not characterize the endopeptidases mutations that correct the morphological defects of the strains lacking 3 or 4 PBPs. It would be a nice addition to show that overexpression of E. coli own endopeptidases does indeed correct the defect of this ∆4 strain, as suggested but not confirmed by this previous study. Such results would strengthens that the role of endopeptidases in correcting carboxypeptidase morphological defects is conserved.

Figure 3, Figure 4, Figure 6B, Supplemental figure 6-10-11: Do the WT and ∆dacA-1 strain harbor the PIPTG-empty insertion at the lacZ locus? Should not these strains be used for proper comparison? Especially since the effects are quite small (see Figure 6B, suppl. Figure 10-11), we should rule out that the insertion might be responsible for the mild effect observed.

Minor points:

Lane 212-213-214: For a broad audience who is not necessarily familiar with the in vivo PG analysis, the abbreviations M5-D44-D45 should be explained or illustrated.

Lane 224: “The mutations in MurA and MurC…” MurA should be replaced by MurD.

Chapter starting at lane 313: Please, keep the same name all along the text for the E. coli strain: ∆4 or∆4Ec.

Lane 452: flanking instead of flaking primers.

Lane 531: SF, probably Salt Free but the abbreviation has not been introduced previously.

Lane 563: enolpyruvate instead of enolpyrupate.

Lane 565: Precise (Fig. 5C).

Lane 745: two-way ANOVA instead of 2way ANOVA.

Lane 761-762:Domain 1 and Domain 3 instead of Dom1 and Dom3.

Lane 769-770: Please, adjust the italic and add “with”: “replacements of murA and murD with murAP122S, murAL35F and murDD447E, respectively”.

Lane 791: “1 mM or 0.2% glucose”, IPTG is missing.

Figure 3A: Like previously, does SF stand for Salt Free? If yes, please introduce the abbreviation somewhere. All the end of the legend can be removed as it is not relevant for this figure.

Figure 3B: Please, indicate the size represented by the bar.

Figure 5E: Please, increase the size of the text under the size bar.

Supplemental figure 1 legend: MicrobeJ instead of MicobeJ.

Supplemental figure 3 legend lane 8: 1 mM IPTG or (instead of ot) 0.2% glucose.

Supplemental figure 4: What the role of the GSK motif? It was never introduced.

Supplemental figure 6 legend: MurCA132T instead of MurAA132T. Modify the name of the strain in the figure: WT PIPTG-MurCA132T instead of MurCA123T.

**Have all data underlying the figures and results presented in the manuscript been provided?**

Reviewer #1: Yes

Reviewer #2: Yes

Reviewer #3: Yes

PLOS authors have the option to publish the peer review history of their article (what does this mean?). If published, this will include your full peer review and any attached files.

Reviewer #1: No

Reviewer #2: No

Reviewer #3: No

---

## [Decision Letter · Decision Letter 1]

10 Mar 2024

Dear Dr Dörr,

Thank you very much for submitting your revised Research Article entitled 'Genetic interaction mapping reveals functional relationships between peptidoglycan endopeptidases and carboxypeptidases.' to PLOS Genetics.

The revised manuscript was fully evaluated at the editorial level and by independent peer reviewers. In general the referees were happy with the revisions. However, Referee #1 still has some concerns regarding your model. Specifically, I ask you to better discuss the possibility that the interaction can be explained by the salt sensitivity of ShyA. At this point I do not ask for additional experiments, but please address the remaining comments in a revised manuscript.

Yours sincerely,

Jan-Willem Veening, Ph.D.

Academic Editor

PLOS Genetics

Lotte Søgaard-Andersen

Section Editor

PLOS Genetics

Reviewer's Responses to Questions

**Comments to the Authors:**

Reviewer #1: The authors have dealt with most of my comments in a satisfactory way. Nevertheless,

my original main criticism still stands – the authors claim a genetic interaction between DacA1 and Endopeptidases – notably ShyA. I think that the 'interaction' is based on environmental conditions. In my view, the interaction can be explained by the salt sensitivity of ShyA. This is not something that should be explored in a separate paper - especially since the authors also speculate on the role of the SMF in the observed phenotypes.

High salt is known to increase hydrophobic interactions – a disruption of such interactions, as in the L109K mutant – activates ShyA. Thus, at low salt, ShyA is more active. ShyA prefers tetra- over pentapeptide PG (earlier work by the authors). Thus, it would follow that the less active ShyA at high salt cannot sufficiently process peptidoglycan when DacA1 is absent – because of the increase in pentapeptide PG, whereas at low salt ShyA is active enough to sufficiently process PG in the absence of DacA1. The hyperactive ShyA then is only tolerated when there is more pentapeptide PG since that is cleaved less efficiently. All the other suppressor mutations also point to this – increases in pentapeptide PG are compensated for by downregulation of PG synthesis or upregulation of PG degradation.

The authors speculate on a role for the SMF – while providing some data that hints in this direction, this is far from conclusive, and in my view this should be considered together with the effects of salt on ShyA activity.

Reviewer #2: The authors have addressed or provided reasonable explanation to all of my previous concerns. In my opinion the revised paper has significantly improved during the review process and in its current form would be of great interest to scientific community and thus should be recommend for publication.

Reviewer #3: The revised manuscript has addressed my main comments and most of those of the other reviewers.

With a few minor corrections listed below, I believe the manuscript can be accepted.

Minor comments:

Lines 148, 526, 546, 835: More a suggestion, but for Remazol Brilliant Blue (RBB), keep the same name/abbreviation along the manuscript, not Remazol brilliant or Remazol-Blue.

Line 225: murAP122S instead of murA P1222

Line 280: MurC instead of fMurC

Line 292: Please verify if any letters are appearing as barred in the PDF (e.g.: partially)

Line 580: The sentence is incomplete: “Genomic DNA from was extracted…”.

Line 603: anti- αRpoA instead of anti- αRpoI

Line 633: acetylglucosamine instead of “acetylglucodamine”

Legends of supplemental figures:

Lines 924, 940, 942, 955 : A) and B), etc.. in bold

Line 956: remove the second B)

**Have all data underlying the figures and results presented in the manuscript been provided?**

Reviewer #1: **No: **Tn seq experiments generate lots of data. I'm not sure what PLoS policies are on these types of data but access to the data may help other researchers.

Reviewer #2: Yes

Reviewer #3: Yes

PLOS authors have the option to publish the peer review history of their article (what does this mean?). If published, this will include your full peer review and any attached files.

Reviewer #1: No

Reviewer #2: No

Reviewer #3: No

---

## [Editor Report · Decision Letter 2]

25 Mar 2024

Dear Dr Dörr,

We are pleased to inform you that your manuscript entitled "Genetic interaction mapping reveals functional relationships between peptidoglycan endopeptidases and carboxypeptidases." has been editorially accepted for publication in PLOS Genetics. Congratulations!

Yours sincerely,

Jan-Willem Veening, Ph.D.

Academic Editor

PLOS Genetics

Lotte Søgaard-Andersen

Section Editor

PLOS Genetics

Comments from the reviewers (if applicable):

**Data Deposition**

http://datadryad.org/submit?journalID=pgenetics&manu=PGENETICS-D-23-01149R2

**Press Queries**

---

## [Editor Report · Acceptance letter]

3 Apr 2024

PGENETICS-D-23-01149R2 

Genetic interaction mapping reveals functional relationships between peptidoglycan endopeptidases and carboxypeptidases. 

Dear Dr Dörr, 

We are pleased to inform you that your manuscript entitled "Genetic interaction mapping reveals functional relationships between peptidoglycan endopeptidases and carboxypeptidases." has been formally accepted for publication in PLOS Genetics! Your manuscript is now with our production department and you will be notified of the publication date in due course.

With kind regards,

Anita Estes

PLOS Genetics

On behalf of:
